# Margin-Independent Online Multiclass Learning via Convex Geometry

**Guru Guruganesh**
Google Research
gurug@google.com

**Allen Liu**
MIT
cliu568@mit.edu

**Jon Schneider**
Google Research
jschnei@google.com

**Joshua Wang**
Google Research
joshuawang@google.com

## Abstract

We consider the problem of multi-class classification, where a stream of adversarially chosen queries arrive and must be assigned a label online. Unlike traditional bounds which seek to minimize the misclassification rate, we minimize the total distance from each query to the region corresponding to its correct label. When the true labels are determined via a nearest neighbor partition – i.e. the label of a point is given by which of $k$ centers it is closest to in Euclidean distance – we show that one can achieve a loss that is independent of the total number of queries. We complement this result by showing that learning general convex sets requires an almost linear loss per query. Our results build off of regret guarantees for the geometric problem of contextual search. In addition, we develop a novel reduction technique from multiclass classification to binary classification which may be of independent interest.

## 1 Introduction

Online multiclass classification is a ubiquitous problem in machine learning. In this problem, a learning algorithm is presented with a stream of incoming query points and is tasked with assigning each query with a label from a fixed set. After choosing a label, the algorithm is told the true label of the query point. The goal of the algorithm is to learn over time how to label the query points as accurately as possible.

Traditionally, theoretical treatments of this problem are built around the notion of a *margin* $\gamma$. This margin represents the extent to which the input points are well-separated from the boundaries between different labels. For example, the analysis of the classic Perceptron algorithm [Nov63] guarantees that it makes at most $O(1/\gamma^2)$ mistakes when performing binary classification, as long as all query points are distance at least $\gamma$ from a hyperplane separating the two classes. More sophisticated analyses and algorithms (relying on e.g. hinge loss) do not necessarily assume the classes are as well separated, but still inherently incorporate a margin $\gamma$ (for example, the hinge loss associated with a point is positive unless it is $\gamma$-separated).

In this paper, we present an alternative to the traditional margin approaches. Our approach weights each mistake by how ambiguous the classification task is for that point, rather than penalizing all mistakes equally. More precisely, consider a partition of the space of all possible query points into $k$ regions $R_i$, where $R_i$ contains all query points whose true label is $i$. In our formulation assigning a query point $q$ a label $i$ incurs a loss of $\ell(q, R_i)$, where $\ell(q, R_i)$ should be thought of as *the distance needed to move $q$ so that it lies in $R_i$* (i.e., for it to be labelled correctly). For example, in the case of

35th Conference on Neural Information Processing Systems (NeurIPS 2021).

a linear classifier, $\ell(q, R_i)$ is zero if $q$ is correctly classified, and the distance to the classifier if $q$ is incorrectly classified. The goal of the algorithm is to minimize the total loss.

This notion of loss not only measures the rate of errors but also the degree of each error; choosing a wildly inaccurate label is punished more than selecting a label that is "almost" correct. This fine-grained approach to looking at errors has occurred in other areas of machine learning research as well. For example, the technique of knowledge distillation is based on training a smaller model on the logits produced by a larger model [HVD15]. Hinton et al. explain, "The relative probabilities of incorrect answers tell us a lot about how the cumbersome model tends to generalize. An image of a BMW, for example, may only have a very small chance of being mistaken for a garbage truck, but that mistake is still many times more probable than mistaking it for a carrot." Rather than leaning on the power of a trained model, our framework differentiates between these different incorrect answers based on the geometry of the problem.

## 1.1 Our Results

### 1.1.1 Learning Linear Classifiers

In this case we have a binary classification problem, where the two regions $R_1$ and $R_2$ are separated by an unknown $d$-dimensional hyperplane $\langle v, x \rangle = 0$ (with $||v||_2 = 1$). Our loss function in this case is the function $\ell(q, R_i) = |\langle q, v \rangle| \cdot \mathbf{1}(q \notin R_i)$. We prove the following result:

**Theorem 1** (Restatement of Corollary 3.1). *There exists an efficient algorithm for learning a linear classifier that incurs a total loss of at most $O(d \log d)$.*

Note that the total loss in Theorem 1 is *independent* of the time horizon (number of rounds) $T$. More importantly, note that this is stronger than the naive guarantee implied by the margin bounds for the Perceptron algorithm. Indeed, each mistake at a distance $\gamma$ from the separating hyperplane is assigned loss $O(\gamma)$ in our model. Since the Perceptron algorithm can make up to $O(1/\gamma^2)$ such mistakes, this only implies Perceptron incurs a total loss of at most $O(1/\gamma)$ (which blows up as $\gamma \to 0$).

Indeed, our algorithm in Theorem 1 is not based off of the Perceptron algorithm or its relatives, but rather off of recently developed algorithms for a problem in online learning known as *contextual search*. In contextual search, a learner is similarly faced with a stream of incoming query points $q_t$ and wishes to learn a hidden vector $v$. However, instead of trying to predict the sign of $\langle v, q_t \rangle$, in contextual search the goal is to guess the *value* of $\langle v, q_t \rangle$. After the learner submits a guess, they are told whether or not their guess was higher or lower than the true value of $\langle v, q_t \rangle$ (and pay a loss equal to the distance between their guess and the truth). The best known contextual search algorithms rely on techniques from integral geometry (bounding various intrinsic volumes of the allowable knowledge set), and are inherently different than existing Perceptron/SVM-style algorithms.

While it may seem like contextual search (which must predict the value of $\langle v, q_t \rangle$ instead of just the sign) is strictly harder than our online binary classification problem, they are somewhat incomparable (for example, unlike in contextual search, we have no control over what feedback we learn about the hidden vector $v$). Nonetheless, in Theorem 8 we show a general reduction from our binary classification problem to contextual search. This allows us to use recent results of [LLS20] to obtain our $O(d \log d)$ bound in Theorem 1.

### 1.1.2 Learning Nearest Neighbor Partitions

One natural way to split a query space into multiple classes is via a *nearest neighbor partition*. In this setting, each label class $i$ is associated with a "center" $x_i \in \mathbb{R}^d$, and each region $R_i$ consists of the points which are "nearest" to $x_i$. To define "nearest", we introduce a similarity metric $\delta(x, y)$ representing the "distance" between points $x$ and $y$ in $\mathbb{R}^d$. The two major classes of similarity metrics we consider are: a) the *inner-product similarity* $\delta(x, y) = -\langle x, y \rangle$ and b) the $L^p$ *similarity* $\delta(x, y) = ||x - y||_p$. Given a fixed similarity metric $\delta$, our loss function in this case is the function $\ell(q, R_i) = \delta(q, x_i) - \min_{i^*} \delta(q, x_{i^*})$; in other words, the difference between the similarity between $q$ and $x_i$ with the similarity between $q$ and its most similar center.

**Theorem 2** (Restatement of Corollary 3.3). *For inner-product similarity, there exists an efficient randomized algorithm for learning a nearest neighbors partition that incurs a total expected loss of at most $O(k^2 d \log d)$.*

Like some other algorithms for multiclass classification, our algorithm in Theorem 2 works by running one instance of our binary classification algorithm (Theorem 1) for each pair of labels. Unlike some other "all-vs-all" methods in the multiclass classification literature, however, it does not suffice to run a simple majority vote over these instances. Instead, to prove Theorem 2, we solve a linear program to construct a probability distribution over centers that guarantees that our expected loss is bounded by an expected decrease in a total potential of all our $\binom{k}{2}$ sub-algorithms.

Our results for $L^p$ similarity are as follows:

**Theorem 3** (Restatement of Theorems 12 and 13). *For $L^p$ similarity, when $p$ is a positive even integer, there exists an efficient randomized algorithm for learning a nearest neighbors partition that incurs a total expected loss of at most $O(k^2\mathrm{poly}(p,d))$.*

*For an arbitrary $p \geq 2$, if all $k$ centers are $\Delta$-separated in $L^p$ distance, there exists an efficient randomized algorithm that incurs a total expected loss of*

$$\frac{k^2\mathrm{poly}(p,d)}{\Delta} \cdot \left(\frac{1}{p-2}\right)^2 .$$

When $p$ is an even integer, it is possible to construct a polynomial kernel that exactly reduces this problem to the problem for inner-product similarity (albeit in the higher dimensional space $\mathbb{R}^{d(p+1)}$). When $p$ is not an even integer, it is no longer possible to perform an exact reduction to the inner-product similarity problem. Instead, in parallel we perform a series of approximate reductions to inner-product similarity at multiple different scales (the full algorithm can be found in Appendix[1] E). Surprisingly, this technique only gives $T$-independent bounds on the loss when $p \geq 2$. It is an interesting open problem to develop algorithms for the case $1 \leq p < 2$ (and more generally, for arbitrary norms).

### 1.1.3 Learning General Convex Regions

Finally, we consider the case where the regions $R_i$ are not defined in relation to hidden centers, but where they can be any convex subsets of $\mathbb{R}^d$. Interestingly, in this case it is impossible to achieve total loss independent of $T$. Indeed, we prove a lower bound of $\Omega(T^{1-O(1/d)})$ for the total loss of any algorithm for this setting, even when $k = 2$.

**Theorem 4.** *Any algorithm for learning general convex regions incurs a total loss of at least $\Omega\left(T^{(d-4)/(d-2)}\right)$, even when there are only two regions.*

### 1.2 Mistake Bounds and Halving Algorithms

As mentioned in the introduction, classical algorithms for multi-class classification generally try to minimize the *mistake bound* (the total number of classification errors the algorithm makes) under some margin guarantee $\gamma$. Even though our algorithms are designed to minimize the absolute loss and not a margin-dependent mistake bound, it is natural to ask whether our algorithms come with any natural mistake bound guarantees.

We show that our algorithms do in fact possess strong mistake bounds, matching the dependence on the margin $\gamma$ of the best known halving algorithms. In particular, we show the following.

**Theorem 5** (Restatement of Theorem 16). *If all query points $q_t$ are at least distance $\gamma$ away from the separating hyperplane, our algorithm for learning linear classifiers (Theorem 1) makes at most $O(d\log(d/\gamma))$ mistakes.*

**Theorem 6** (Restatement of Theorem 17). *If all query points $q_t$ are at least distance $\gamma$ away from the boundary between any two regions, our algorithm for learning nearest neighbor partitions (Theorem 2) makes at most $O(k^2 d\log(d/\gamma))$ mistakes.*

In comparison, the best dimension-dependent classical bounds for this problem come from halving algorithms (efficiently implementable via linear programming) which have mistake bounds of $O(d\log(1/\gamma))$ and $O(kd\log(1/\gamma))$ respectively. In the first case, our mistake bound is nearly tight (losing only an additive $O(d\log d)$). In the second case, our mistake bound is tight up to a multiplicative factor of $k$; it is an interesting open question whether it is possible to remove this factor of $k$ in our techniques.

---

[1]All references to the appendix refer to the appendices of the Supplementary Material.

We additionally introduce a variant on the mistake bound that we call a *robust mistake bound* that is defined as follows. Normally, when we have a margin constraint of $\gamma$, we insist that all query points $q_t$ are distance at least $\gamma$ away from any separating hyperplane. In our robust model, we remove this constraint, but only count mistakes when the query point $q_t$ lies at least $\gamma$ away from the separating hyperplane.

Existing algorithms (both the Perceptron and halving algorithms) do not appear to give any non-trivial mistake bounds in the robust model – it is very important for the analysis of these algorithms that *every* query point is far from the separating hyperplane. On the other hand, it straightforwardly follows from our notion of loss that if we have an $O(R)$-loss algorithm for some problem, that algorithm simultaneously achieves an $O(R/\gamma)$ robust mistake bound. In particular, we obtain robust mistake bounds of $O((d \log d)/\gamma)$ and $O((k^2 d \log d)/\gamma)$ for learning linear classifiers and learning nearest neighbor partitions respectively.

**Related work**   The problem of online binary classification (and specifically, of online learning of a linear classifier) is one of the oldest problems in machine learning. The Perceptron algorithm was invented in [Ros58], and the first mistake bound analysis of the Perceptron algorithm appeared in [Nov63]. Since then, there has been a myriad of research on this problem, some of which is well-surveyed in [MR13]. Of note, the first mistake bounds for the non-separable case appear in [FS99]. We are not aware of any work on this problem that investigates the same loss we present in this paper.

Bounds for support vector machines (see [Vap13]) also result in the use of a margin to bound the number of mistakes. Choosing a suitable kernel can help create or improve the margin when viewing the data in the "kernel" space. We use a similar technique in proving bounds for generalized $L^p$ norms. Our idea is to use differently-scaled kernels to produce increasingly accurate approximations. To the best of our knowledge, the technique we present is novel. There are other related techniques in the literature, e.g. [SBD06] attempt to learn the best kernel to improve classification error.

Similarly, there is a wealth of both theoretical and empirical research on multiclass classification. As far back as 1973, researchers were looking at generalizing binary classification algorithms to this setting (see e.g. Kesler's construction in [DH+73]). One piece of relevant work is [CS03], which generalizes online binary classification Perceptron-style algorithms to solve online multiclass classification problems – in particular, the multiclass hypotheses they consider are same as the nearest neighbor partitions generated by the inner-product similarity metric (although as with the Perceptron, they only analyze the raw classification error). Since then, this work has been extended in many ways to a variety of settings (e.g. [CDK+06, CG13, KSST08]).

Another way of looking at the problem of multiclass classification is that we are learning how to cluster a sequence of input points into $k$ pre-existing clusters. Indeed, the nearest neighbor partition with $L^2$ similarity metric gives rise to exactly the same partitioning as a $k$-means clustering. There is an active area of research on learning how to cluster in an online fashion (e.g. [GLZ14, GLK+17, BR20]). Perhaps most relevantly, [LSS16] studies a setting where one must choose cluster labels for incoming points in an online fashion, and then at the end of the algorithm can choose the $k$ centers (the goal being to minimize the total $k$-means loss of this eventual clustering).

The algorithms we develop in this paper are based off of algorithms for contextual search. Contextual search is a problem that originally arose in the online pricing community [CLPL16, LLV17]. The variant of contextual search we reference in this paper (with symmetric, absolute value loss) first appeared in [LS18]. The algorithms in this paper were later improved in [LLS20] (and it is these improved algorithms that we build off of in this paper).

## 2   Model

**Notation**   We will write $B_d(c, r)$ to denote the $d$-dimensional ball in $\mathbb{R}^d$ centered at $c$ with radius $r$. We write $B_d$ in place of $B_d(0, 1)$ to indicate the unit ball centered at the origin.

Given an $x \in \mathbb{R}^d$, we will write $\|x\|_p$ to denote the $L^p$ norm of the vector $x$. In the case of $p = 2$, we will often omit the subscript and write $\|x\|$ in place of $\|x\|_2$.

## 2.1 Online Multiclass Learning

We will view the problem of online multiclass learning as follows. There are $k$ disjoint regions in some domain, say $B_d$, labelled $R_1$ through $R_k$. The region $R_i$ contains the points in $B_d$ that should be assigned the label $i$. The goal of the learner is to learn these subsets (and thus how to label points in $B_d$ in an online manner). Every round $t$, the learner receives an adversarially chosen query point $q_t \in B_d$. The learner must submit a prediction $I_t \in [k]$ for which region $R_{I_t}$ the point $q_t$ lies in. The learner then learns which region $R_{I_t^*}$ the point actually belongs to, and suffers some loss $\ell(q_t, R_{I_t})$. This loss function should in some way represent how far $q_t$ was from lying in the region $R_{I_t}$ chosen by the learner; for example, in the case where the learner chooses the correct region $R_{I_t^*}$, $\ell(q_t, R_{I_t^*})$ should be zero.

In this paper, we will consider two specific cases of the above learning problem. In the first case, we wish to learn a *nearest-neighbor partition*. That is, the $k$ regions are defined by $k$ "centers" $x_1, x_2, \ldots, x_k \in B_d$. Region $R_i$ then consists of all the points which are "nearest" to center $x_i$ according to some similarity metric $\delta(x, y)$ (where lower values of $\delta(x, y)$ mean that $x$ and $y$ are more similar; note that $\delta(x, y)$ *does not* need to be an actual metric obeying the triangle-inequality). Given a similarity metric $\delta(x, y)$, the loss our algorithm incurs when labelling query $q$ with label $i$ is given by $\ell(q, R_i) = \delta(q, x_i) - \delta(q, x_{i^*})$, where $R_{i^*}$ is the region containing query $q$.

We will examine several different possibilities for $\delta(x, y)$, including:

- $\delta(x, y) = -\langle x, y \rangle$. We refer to this as the *inner-product similarity* between $x$ and $y$. Note that when $k = 2$, using this similarity metric reduces to the problem of learning a linear classifier. For $k > 2$, this results in similar partitions to those learned by multiclass perceptrons / SVMs [CS03, CDK$^+$06].
- $\delta(x, y) = ||x - y||_2$; in other words, the Euclidean distance between $x$ and $y$. When using this loss function, the $k$ regions are given by the Voronoi diagram formed by the $k$ centers $x_i$.
- For $p \geq 1$, $\delta(x, y) = ||x - y||_p$; in other words, the $L^p$ distance between $x$ and $y$.

There is a straightforward reduction from the Euclidean distance similarity to the inner-product similarity (see Appendix C), so in Section 3 we will primarily concern ourselves with the inner-product similarity. In Section 4 we will tackle this problem for the case of general $L^p$ norms; for some cases (even integer $p$) it is possible to perform a similar reduction to the inner product similarity, but in general it is not and we will need to rely on other techniques.

In the second case, we wish to learn *general convex sets*. In particular, there are $k$ disjoint convex sets $R_1, \ldots, R_k \subseteq B_d$ (not necessarily a partition). Each round $t$, we receive a query point $q_t \in \bigcup_{i=1}^k R_i$, guess a region $i \in [k]$, and are penalized against the loss function $\ell(q_t, R_i) \triangleq \min_{x \in R_i} ||q_t - x||_2$. In other words, we are penalized the minimum distance from $q_t$ to the predicted region $R_i$, which is zero if our guess was correct. In Section 5 we will show that (even in the case of $k = 2$), there is no low-loss learning algorithm for learning general convex sets (in contrast to learning nearest neighbors).

Finally, for any $\alpha > 0$ and existing loss function $\ell(g, R_i)$, we can consider the modified loss function $\ell'(q, R_i) = \ell(q, R_i)^\alpha$. Note that this does not change the division of $B_d$ into regions, but it *can* change the total loss incurred by our algorithm (and will be useful for some of our reductions). If in any result we do not specify an $\alpha$, that means we are taking $\alpha = 1$ (i.e. the unmodified loss function).

## 2.2 Contextual Search

One of the main tools we will rely on is an existing algorithm for a problem in online learning known as *contextual search*. For our purposes, the problem of contextual search can be defined as follows. There is a hidden point $p \in B_d$, unknown to the learner. Every round $t$ (for $T$ rounds) an adversary provides the learner with a query vector $x_t \in B_d$. The learner must then submit a guess $g_t$ for the value of the inner product $\langle x_t, p \rangle$. The learner then learns whether their guess was too high or too low. At the end of the game, the learner incurs loss $\ell(g_t, \langle x_t, p \rangle) = |g_t - \langle x_t, p \rangle|$ for each of their guesses. The learner's goal is to minimize their total loss.

Interestingly, there exist algorithms for contextual search with total loss polynomial in the ambient dimension $d$ and independent of the time horizon $T$. The first such algorithms were given in [LS18]

and used techniques from integral geometry to obtain a total regret of $O(d^4)$. More recently, these algorithms were improved in [LLS20] to achieve a regret bound of $O(d \log d)$. We will rely on a slightly strengthened variant of the result from [LLS20] to work when the loss function raised to an arbitrary power.

**Theorem 7.** *Let $\alpha > 0$. There exists an algorithm for contextual search with loss function $\ell(g_t, \langle x_t, p \rangle) = |g_t - \langle x_t, p \rangle|^\alpha$ that incurs a total loss of at most $O(\alpha^{-2} d \log d)$.*

In particular, Theorem 7 is satisfied by the algorithm from [LLS20]. For completeness, we include a description of the algorithm (along with the proof of Theorem 7) in Appendix A.

## 3 Learning Nearest Neighbor Partitions

### 3.1 The Two-Point Case: Learning a Hyperplane

To begin, we will discuss how to solve the $k = 2$ variant of the problem of learning nearest neighbor partitions for the inner-product similarity function $\delta(q, x) = - \langle q, x \rangle$. Recall that in this setting we have two unknown centers $x_1$ and $x_2$ belonging to $B_d$. Each round $t$ we are given a query point $q_t \in B_d$, and asked to choose a label $I_t \in \{1, 2\}$ of the center that we think is most similar to $q_t$ (i.e., that maximizes $\delta(q_t, x)$). Letting $y_t = x_{I_t}$ and $x_t^* = \arg\max_{x_i} \delta(q, x_i)$, our loss in round $t$ is zero if we guess correctly ($y_t = x_t^*$) and is $\delta(q, x_t^*) - \delta(q, y_t)$ if we guess incorrectly (we will also be able to deal with the case when the loss is $|\delta(q, x_t^*) - \delta(q, y_t)|^\alpha$ for some $\alpha > 0$). Afterwards, we are told the identity (but not the location) of $x_t^*$.

Note that the optimal strategy in this game (for an agent who knows the hidden centers $x_1$ and $x_2$) is to guess $I_t = 1$ whenever $\langle q, x_1 \rangle > \langle q, x_2 \rangle$, and to guess $I_t = 2$ otherwise. Rewriting this, we want to guess $I_t = 1$ exactly when $\langle q, x_1 - x_2 \rangle > 0$. If we let $w = x_1 - x_2$, we can think of goal as learning the hyperplane $\langle q, w \rangle = 0$. More specifically, each round we are given a point $q$ and asked which side of the hyperplane $q$ lies on. If we guess correctly, we suffer no loss; if we guess incorrectly, we suffer loss equal to the distance from $q$ to this hyperplane. In either case, we learn afterwards which side of the hyperplane $q$ lies on.

#### 3.1.1 Reducing to Contextual Search

We will show that we can solve this online learning problem with total loss $O(\text{poly}(d))$, independent of the number of rounds $T$ in the time-horizon. As mentioned earlier, our primary tool will be existing algorithms for a problem in online learning known as *contextual search*.

Recall that in contextual search there is a hidden vector $v \in B_d$, and each round we are given a vector $q_t \in B_d$. However, unlike in our problem (where we only care about the sign of the inner product $\langle q_t, v \rangle$), the goal in contextual search is to submit a guess $g_t$ for the value of the inner product $\langle q_t, v \rangle$. We then incur loss equal to the absolute distance $|\langle q_t, v_t \rangle - g_t|$ between our guess and the truth, and are then told whether our guess $g_t$ was too high or too low.

As mentioned earlier in Section 2.2, there exist algorithms for contextual search with $O(d \log d)$ total loss. Via a simple reduction, we will show that we can apply these algorithms in our setting.

**Theorem 8.** *Fix $\alpha > 0$. Assume there exists an algorithm $\mathcal{A}$ for contextual search with loss function $\ell(g_t, \langle x_t, p \rangle) = |g_t - \langle x_t, p \rangle|^\alpha$ that incurs regret at most $R(d, T)$. Then there exists an algorithm $\mathcal{A}'$ that incurs regret at most $R(d, T)$ for the $k = 2$ case of learning nearest neighbor partitions with similarity metric $\delta(x, y) = - \langle x, y \rangle$ and loss raised to the power $\alpha$.*

**Corollary 3.1.** *Fix an $\alpha > 0$. When $k = 2$, there exists an algorithm for learning nearest neighbor partitions with similarity metric $\delta(x, y) = - \langle x, y \rangle$ and loss raised to the power $\alpha$ that incurs total loss at most $O(\alpha^{-2} d \log d)$.*

#### 3.1.2 Potential-Based Algorithms

In order to generalize this to $k > 2$ labels, we will need to open the black box that is our contextual search algorithm slightly. In particular, the argument in the following section requires our algorithm for the $k = 2$ case to be a *potential-based algorithm*.

Before defining exactly what a potential-based algorithm is, it will be useful to define the notion of a *knowledge set*. For the problem we are considering in this section – the $k = 2$ variant of our problem

for inner-product similarity – we will define the *knowledge set* $K_t$ at time $t$ to be the set of possible values for $w = x_1 - x_2$ that are consistent with all known information thus far. Note that since $x_1$ and $x_2$ start as arbitrary points in $B_d$, the knowledge set $K_0$ is simply the set $B_d - B_d = 2B_d$. As the algorithm gets more feedback about $w$, the knowledge set shrinks; however, since this feedback is always of the form of a linear constraint (e.g. $\langle q_t, w \rangle \geq 0$), the knowledge set $K_t$ is always a convex subset of $\mathbb{R}^d$.

Let $S_t$ be the history of all feedback the algorithm has seen up to (but not including) round $t$; that is $S_t = \{(q_1, I_1^*), (q_2, I_2^*), \ldots, (q_{t-1}, I_{t-1}^*)\}$. Let $\mathcal{S}_t$ denote the set of possible values for $S_t$, and let $\mathcal{S} = \bigcup_t \mathcal{S}_t$. We can think of $S_t$ as capturing the *state* of a deterministic algorithm at time $t$. For now, it is fine to think of $S_t$ as interchangeable with $K_t$; i.e., the knowledge set $K_t$ captures all relevant details about all feedback the algorithm has observed thus far. (Later, when looking at $L^p$ similarity metrics, we will want to keep track of separate knowledge sets at different scales, and thus will want a more nuanced notion of potential-based algorithm).

**Definition 9.** *A deterministic algorithm $\mathcal{A}$ (for $k = 2$ and $\alpha > 0$) is a potential-based algorithm if there exists a potential function $\Phi$ from $\mathcal{S}$ to $\mathbb{R}_{\geq 0}$ and a "loss bound" function $L$ from $\mathcal{S} \times B_d$ to $\mathbb{R}_{\geq 0}$ that satisfy:*

- *For all rounds $t$, $\Phi(S_{t+1}) \leq \Phi(S_t)$.*
- *Let $q_t$ be the query point in round $t$. Then $L(S_t, q_t)$ is an upper bound on the loss incurred by **any guess**. In other words, $L(S_t, q_t)$ must satisfy*

$$L(S_t, q_t) \geq \max_{\substack{x_1, x_2 \\ \text{consistent with } S_t}} |\delta(q_t, x_1) - \delta(q_t, x_2)|^\alpha .$$

- *Again, let $q_t$ be the query point in round $t$. If $\mathcal{A}$ guesses the label incorrectly in round $t$, then $\Phi(S_t) - \Phi(S_{t+1}) \geq L(S_t, q_t)$.*

For the case of inner-product similarity, we will set the loss bound function $L(S_t, q_t)$ equal to the width of the knowledge set $K_t$ in the direction $q_t$. Importantly, this choice of $L$ is an efficiently computable (in terms of $q_t$ and the knowledge set $K_t$) upper-bound on the loss, which will prove important in the following section (in general, we will want both $\Phi$ and $L$ to be efficiently computable in order to efficiently carry out the reduction in Section 3.2).

Note also that such a potential immediately gives a way to bound the total loss of an algorithm independently of $T$; in particular, summing the inequality $\Phi(S_t) - \Phi(S_{t+1}) \geq L(S_t, q_t)$ over all $t$ gives that the total loss is at most $\Phi(S_0)$. We call the value $\Phi(S_0)$ the *initial potential* of the algorithm $\mathcal{A}$.

Similar potential-based arguments are used in [LLS20] and [LS18] to give $T$-independent total loss bounds for the problem of contextual search. Unsurprisingly, these arguments can be extended (via Theorem 8) to apply to the problem of learning nearest neighbors as well.

**Theorem 10.** *Fix an $\alpha > 0$. When $k = 2$, there exists a potential-based algorithm for learning nearest neighbor partitions under the similarity metric $\delta(x, y) = -\langle x, y \rangle$ that incurs total loss at most $O(\alpha^{-2} d \log d)$ (independent of the time horizon $T$).*

The proof of Theorem 10 can be found in Appendix B.

## 3.2 From Two to Many Centers

We will now show how to use any potential-based algorithm for learning nearest neighbor partitions with two centers to construct an algorithm that can learn nearest neighbor partitions with any number of centers.

Our main result is the following:

**Theorem 11.** *Let $\mathcal{A}$ be a potential-based algorithm for learning nearest neighbor partitions with two centers that has an initial potential (and thus a total loss) of at most $R$. Then there exists a randomized algorithm $\mathcal{A}'$ for learning nearest neighbor partitions for any $k \geq 2$ whose total expected loss is at most $O(k^2 R)$.*

Similar to many existing methods for multiclass classification ("all-to-all" methods), we will accomplish this by running one instance of our two-center algorithm $\mathcal{A}$ for each of the $\binom{k}{2}$ pairs of centers.

However, instead of using a simple majority voting scheme to choose our eventual label, we will use the *potentials* of these $\binom{k}{2}$ algorithms to construct a distribution over centers that we will sample from.

More specifically, our algorithm will work as follows. As mentioned, each round, based on the current query $q_t$ and the potentials of the $\binom{k}{2}$ sub-algorithms, we will construct a distribution $v \in \Delta([k])$ over the $k$ centers (we will describe how we do this shortly). We then sample a label $i$ from this distribution $v$ and guess it as the label of $q_t$. If we then learn that the correct label was in fact $j$, we update the sub-algorithm for the pair $(i, j)$ with this information. We do not update any of the other sub-algorithms (in particular, if we guess the label correctly, we do not update any of the sub-algorithms).

To construct our distribution $v$, we will choose a distribution $v$ that has the property that our expected loss in each round is at most the expected decrease in the total potential over all $\binom{k}{2}$ sub-algorithms. This will guarantee that the total expected loss of our algorithm is bounded above by the total starting potential of all our $\binom{k}{2}$ sub-algorithms. To be more precise, define the following variables:

1. Let $\mathcal{A}_{ij}$ denote the two-center sub-algorithm for the labels $i$ and $j$. Let $S_{ij}^{(t)}$ be the state of $\mathcal{A}_{ij}$ at round $t$, and let $\Phi_{ij}^{(t)} = \Phi(S_{ij}^{(t)})$ be the potential of $\mathcal{A}_{ij}$ at round $t$. Define $\Phi^{(t)} = \sum_{(i,j)} \Phi_{ij}^{(t)}$ to be the total of all the potentials belonging to sub-algorithms $\mathcal{A}_{ij}$.

2. As in Definition 9, let $L_{ij}^{(t)} = L(S_{ij}^{(t)}, q_t)$ denote an upper-bound on the loss incurred by the algorithm $\mathcal{A}_{ij}$ in round $t$.

3. Let $D_{ij}^{(t)}$ denote the reduction in the potential of $\mathcal{A}_{ij}$ when $i$ is the correct label. In other words, $D_{ij}^{(t)} = \Phi_{ij}^{(t)} - \Phi_{ij}^{(t+1)}$ when $I_t^* = i$. Note that $D_{ij}^{(t)}$ is *not* equal to $D_{ji}^{(t)}$; it is possible for the potential of $\mathcal{A}_{ij}$ to decrease a lot more upon learning that a point $q_t$ has label $i$ than learning it has label $j$ (geometrically, this corresponds to different halves of the knowledge set being maintained by $\mathcal{A}_{ij}$).

4. Finally, define $M^{(t)} \triangleq D^{(t)} - \frac{1}{2} \cdot L^{(t)}$, where here we are treating $M^{(t)}$ and $L^{(t)}$ as $n$-by-$n$ matrices. Observe that we can efficiently compute the values $L_{ij}^{(t)}$ and $D_{ij}^{(t)}$ and hence the value of $M_{ij}^{(t)}$ from $q_t$ and the knowledge set of $\mathcal{A}_{ij}$ at the beginning of round $t$.

From here on, we will fix a round $t$ and suppress all associated superscripts. Assume that in this round the correct label for the query point is $r$. Now, if we sample a label from a distribution $v$, then note that the expected loss we sustain is $\sum_i L_{ri} v_i = e_r^T L v$. Similarly, the expected decrease in $\Phi$ is $\sum_i D_{ri} v_i = e_r^T D v$. If it was guaranteed to be the case that $e_r^T D v \geq e_r^T L v$, then this would in turn guarantee that our total expected loss is at most the total starting potential.

It follows that if we can find a distribution $v$ that satisfies $Mv \geq 0$, we are in luck. If such a distribution exists, we can find it by solving an LP. This is in fact how we find the distribution $v$, and this concludes the description of the algorithm. To prove correctness of the algorithm, it suffices to show that such a distribution always exists.

To do so, note that from the third point in Definition 9, we know that for each pair of labels $(i, j)$,

$$D_{ij} + D_{ji} \geq L_{ij} \tag{P1}$$

(In fact, Definition 9 tells us that $\max(D_{ij}, D_{ji}) \geq L_{ij}$, since if $\mathcal{A}_{ij}$ predicts $j$, we have that $D_{ij} \geq L_{ij}$, and likewise if $\mathcal{A}_{ij}$ predicts $i$, we have that $D_{ji} \geq L_{ij}$). We can rewrite (P1) in the form $M + M^T \geq 0$. The following lemma shows that if $M + M^T \geq 0$, then there must exist a distribution $v$ satisfying $Mv \geq 0$, whose proof we defer to the appendix.

**Lemma 3.2.** *Given any matrix $M \in \mathbb{R}^{n \times n}$ such that $M + M^T \geq 0$, there exists a point $v \in \Delta_n$ such that $Mv \geq 0$.*

With this, it is straightforward to finish off the proof of Theorem 11.

*Proof of Theorem 11.* By Lemma 3.2, we know that each round we can find a distribution $v \in \Delta([k])$ satisfying $Mv \geq 0$. By the previous discussion, it follows that if we always sample from this

distribution, the total expected loss will be at most twice the starting total potential, $2\Phi^{(1)}$. But note that $\Phi^{(1)}$ is just the sum of the starting potentials $\Phi_{ij}^{(1)}$ of all the sub-algorithms, and is thus at most $\binom{k}{2}R$. It follows that the total loss of our new algorithm is at most $O(k^2R)$. □

**Corollary 3.3.** *Fix an $\alpha > 0$. There exists a randomized algorithm for learning nearest neighbor partitions with the inner-product similarity metric that incurs total loss at most $O(\alpha^{-2}k^2 d \log d)$.*

**Remark 3.4.** *Why do simple algorithms (such as a majority voting scheme) fail to work in our setting? In fact, it is possible to get a simple majority vote (breaking ties arbitrarily) to work if we are given additional feedback from the algorithm – specifically, the ranking of all $k$ distances $\delta(q_t, x_i)$. With this information, it is possible to update all $\binom{k}{2}$ sub-instances each round, and charge any regret we sustain to an appropriate sub-instance. But if we only receive the true label of $q_t$, we no longer have the information to update every sub-instance, and instead have to do the more subtle amortization described above.*

# 4 Learning Nearest Neighbor Partitions Under $L^p$ Similarity

In this section, we will discuss generalizations of our previous results for the inner-product similarity metric to general $L^p$ spaces. We will primarily deal with the case when there are only two unknown points ($k = 2$) as the general reduction in Section 3.2 will allow us to reduce from the $k$-point case to the 2-point case.

The general approach for the algorithms in this section is to apply some sort of kernel mapping so that inequalities of the form $\|X - x_1\|_p \le \|X - x_2\|_p$ become linear constraints. Once we linearize the problem, we can apply our earlier algorithms for the inner-product similarity along with tools from [LLS20].

Similar to the previous sections, we will assume that the hidden points $x_1, x_2$ are in the $L^2$ unit ball $B_d$. This is equivalent to assuming that the hidden points are in the $L^p$ unit ball (which may be a more natural setting since we are working with $L^p$ distances) up to a $\sqrt{d}$ factor since we can simply rescale the $L^2$-ball to contain the $L^p$-ball.

## 4.1 $p$-Norms for Even Integers $p$

When $p$ is an even integer, there is a kernel mapping that exactly linearizes the problem. To see this, note that $|(x - a)|^p = (x - a)^p$ which is a polynomial in $x$ so it suffices to consider the polynomial kernel $(1, x, \ldots, x^p)$. In $d$ dimensions, we can simply apply this kernel map coordinate-wise. After applying these kernel maps, we will be able to apply Corollary 3.3. Our main theorem for even integer $p$ is stated below.

**Theorem 12.** *For even integer $p$, there is an algorithm for learning nearest neighbor partitions under the $L^p$ similarity metric that incurs expected total loss at most $O(p^4 d^{(p+1)/p} k^2 \log d) = O(k^2 \cdot \mathrm{poly}(p, d))$.*

The details of the proof are deferred to Appendix E.2.

**Remark 4.1.** *For the special case of $p = 2$, the reduction is even more immediate (the kernel map needs only add a single dimension), and obtains a slightly tighter bound of $O(k^2 d \log d)$. The reduction for this special case is summarized in Appendix C.*

## 4.2 General $p$-Norms

Now we discuss how to deal with general $L^p$ norms. The main difficulty here is that there is no kernel map that exactly linearizes the problem so instead we will have multiple kernel maps. These kernel maps approximately linearize the problem at different scales, i.e. they have different output dimensions and as the output dimension grows, the problem can be more closely approximated by a linear one. When we are given a query point, we choose the scale of approximation that we use based on estimating the maximum possible loss that we can incur. By balancing the dimensionality and the approximation error to be at the same scale as the maximum possible loss, we may ensure that whenever we guess the label incorrectly, a certain potential function must decrease by an amount comparable to the loss that we incur. Our main theorem is stated below.

**Theorem 13.** *Fix a $p > 2$. If all $k$ unknown centers are $\Delta$-separated in $L^p$ distance, there exists an algorithm for learning nearest neighbor partitions under the $L^p$ similarity metric that incurs total loss*

$$\frac{k^2 \text{poly}(d, p)}{\Delta} \cdot \left(\frac{1}{p-2}\right)^2 .$$

The full proof is more complicated than the algorithms in previous sections and requires opening the contextual search black-box and redoing parts of the analysis. Note that in the above theorem, we need the assumption that the unknown centers are $\Delta$-separated, an assumption that was not necessary for even integer $p$. This is due to the fact that when the true centers are too close together, the dimensionality of the kernels that we need to achieve the necessary approximations are large. Nevertheless, we believe that the separated centers assumption is realistic for classification problems in practice. The details of the proof of Theorem 13 are deferred to Appendix E.3 of the Supplementary Material.

## 5 Learning General Convex Regions: Lower Bound

In this section, we consider the task of learning general convex regions, and present a construction which shows that any learning algorithm incurs $\Omega(T^{(d-4)/(d-2)})$ error over $T$ rounds, even for only $k = 2$ regions. We give the full proof in Appendix F.

**Theorem 4.** *Any algorithm for learning general convex regions incurs a total loss of at least $\Omega\left(T^{(d-4)/(d-2)}\right)$, even when there are only two regions.*

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

## Funding Transparency Statement

Allen Liu is supported by a NSF Graduate Fellowship and a Hertz fellowship. None of the other authors received third-party funding or support during the preparation of this paper.

