# A  Contextual search

In this appendix, we review the contextual search algorithm and analysis presented in [LLS20]. Our presentation will largely follow that of [LLS20], with two minor changes: 1. we will demonstrate that the algorithm works for all loss functions of the form $\ell(g_t, \langle x_t, p \rangle)$, where it incurs total loss at most $O(\alpha^{-2} d \log d)$ (Theorem 7), and 2. we will present the analysis in a slightly different way that makes it easier for us to construct potential-based algorithms for learning nearest neighbors.

The algorithm is outlined in Algorithm 1. Briefly, the algorithm works as follows. Whenever the algorithm gets a query direction $x_t$ from the adversary, the algorithm looks at the width of the current knowledge set in the direction $x_t$. Based on the size of this width, the algorithm picks an "expansion parameter" $z_i$, and chooses a guess $g_t$ so that the hyperplane $\langle v, x_t \rangle = g_t$ splits the volume of $K_t + z_i B_d$ in half.

---

**Algorithm 1** CONTEXTUAL SEARCH ALGORITHM ([LLS20])

---

Initialize $K_1 = B_d$ and $z_i = 2^{-i}/(8d)$ for all $i$.
**for** $t$ in $1, 2, \ldots, T$ **do**
  Adversary picks $x_t$.
  Let $i$ be the largest index such that $\mathsf{width}(K_t; x_t) \leq 2^{-i}$.
  Submit guess $g_t$ such that $\mathsf{Vol}(\{v \in K_t + z_i B_d \mid \langle v, x_t \rangle \geq g_t\}) = \frac{1}{2}\mathsf{Vol}(K_t + z_i B_d)$.
  Update $K_{t+1}$ based on feedback.

---

**Theorem 14** (Restatement of Theorem 7). *Let $\alpha > 0$. Algorithm 1 is an algorithm for contextual search with loss function $\ell(g_t, \langle x_t, p \rangle) = |g_t - \langle x_t, p \rangle|^\alpha$ that incurs a total loss of at most $O(\alpha^{-2} d \log d)$.*

*Proof.* To prove Theorem 7, we will examine the following potential function of the knowledge set at time $t$.

$$\Phi(K_t) = \sum_{i=1}^\infty 2^{-\alpha i} \log \frac{\mathsf{Vol}\,(K_t + z_i B_d)}{\mathsf{Vol}(z_i B_d)}.$$

Our goal will be to show that $\Phi(K_t)$ decreases by at least the loss we sustain in each round. This will bound the total loss Algorithm 1 sustains by at most $\Phi(K_0)$. To do this, we will employ the following lemma from [LLS20]:

**Lemma A.1** (Lemma 2.1 in [LLS20]). *If $i$ is the index chosen at round $t$ in Algorithm 1, then $\mathsf{Vol}(K_{t+1} + z_i B_d) \leq \frac{3}{4}\mathsf{Vol}(K_t + z_i B_d)$.*

Note that Lemma A.1 implies that if $i$ is the index chosen at round $t$, then $\Phi(K_t) - \Phi(K_{t+1}) \geq \left(\log \frac{4}{3}\right) 2^{-\alpha i}$. But also, if $i$ is the chosen index at round $t$, then the width in the query direction is at most $2^{-i}$, and thus the loss sustained in this round is at most $2^{-\alpha i}$. If we let $L_t$ be the loss sustained in round $t$, we have thus shown that

$$\Phi(K_t) - \Phi(K_{t+1}) \geq \left(\log \frac{4}{3}\right) L_t.$$

Summing this over all $t$, we find the total loss is at most $O(\Phi(K_1))$. Since $K_1 = B_d$, we can evaluate $\Phi(K_1)$ as follows:

$$
\begin{aligned}
\Phi(K_1) &= \sum_{i=1}^{\infty} 2^{-\alpha i} \log \frac{\mathsf{Vol}\,(B_d + z_i B_d)}{\mathsf{Vol}(z_i B_d)} \\
&= \sum_{i=1}^{\infty} 2^{-\alpha i} d \log \left( 1 + \frac{1}{z_i} \right) \\
&\leq \sum_{i=1}^{\infty} 2^{-\alpha i} d \log \left( 2^{i+4} d \right) \\
&= O\left( d \sum_{i=1}^{\infty} 2^{-\alpha i} i \right) + O\left( d \log d \sum_{i=1}^{\infty} 2^{-\alpha i} \right) \\
&= O(\alpha^{-2} d) + O(\alpha^{-1} d \log d) \\
&\leq O(\alpha^{-2} d \log d).
\end{aligned}
$$

$\square$

## B   Potential-based algorithm for inner-product similarity

In this appendix we prove Theorem 10, showing that the described algorithm for learning nearest neighbor partitions in Section 3.1 is a potential-based algorithm. Indeed, we will be able to use the same potential function as in the proof of Theorem 14, namely:

$$
\Phi(K_t) = \sum_{i=1}^{\infty} 2^{-\alpha i} \log \frac{\mathsf{Vol}\,(K_t + z_i B_d)}{\mathsf{Vol}(z_i B_d)}.
$$

Note that $\Phi(K_t)$ clearly satisfies the first two conditions in Definition 9, namely $\Phi(K_t) \geq 0$ for any knowledge set $K_t$, and it is always the case that $\Phi(K_{t+1}) \leq \Phi(K_t)$ (in particular, since $K_{t+1} \subseteq K_t$). It thus suffices to show the third condition of Definition 9 holds:

**Lemma B.1.** *Let $\mathcal{A}$ be the $(k = 2)$ algorithm for learning nearest neighbor partitions with similarity metric $\delta(x,y) = -\langle x, y \rangle$ and loss raised to the power $\alpha$ described in Corollary 3.1. Then, for all rounds $t$ where $\mathcal{A}$ guesses the label incorrectly, $\Phi(K_t) - \Phi(K_{t+1}) \geq \Omega\left( \max_{v \in K_t} |\langle q_t, v \rangle|^{\alpha} \right)$.*

*Proof.* Recall from the proof of Theorem 8, whenever $\mathcal{A}$ guesses the label incorrectly, we update the state of the contextual search algorithm underlying $\mathcal{A}$. The contextual search algorithm underlying $\mathcal{A}$ shares the same knowledge set $K_t$ as $\mathcal{A}$, and by the analysis in the proof of Theorem 14, $\Phi(K_t)$ must then satisfy

$$
\Phi(K_t) - \Phi(K_{t+1}) \geq \left( \log \frac{4}{3} \right) \mathsf{width}(K_t; q_t)^{\alpha}.
$$

Since $\mathsf{width}(K_t; q_t) = \max_{v \in K_t} \langle v, q_t \rangle - \min_{v \in K_t} \langle v, q_t \rangle$, $\mathsf{width}(K_t; q_t) \geq \max_{v \in K_t} |\langle q_t, v \rangle|$, and we have proved this lemma. $\square$

## C   From Euclidean Distance to Inner-Product Similarity

We have presented two different variants of the nearest neighbor partition problem: one where we want to return the point $x_i$ with largest inner-product similarity to each query $q_t$, and one where we want to return the point $x_i$ closest to the query $q_t$ in some $L^p$ norm. Here we will show that in the case of the Euclidean norm, we can easily reduce the second problem to the first – and therefore, it suffices to solve the problem only for the case of inner-product similarity in Section 3. More specifically, we will show that if we can solve the nearest neighbor partition problem for the similarity metric $\delta(x,y) = -\langle x, y \rangle$ and $\alpha = 1/2$, we can solve the nearest neighbor partition problem for the $L^2$ similarity metric $\delta(x,y) = \|x - y\|_2$. (In many ways, this can be seen as a warm-up for the more general case of even integer $p$ $L^p$ norms in Appendix E.2).

Let $x_1, x_2, \ldots, x_k$ be points in $B_d$. Consider following two maps $T$ and $Q$ from $B_d \to B_{d+1}$. $T$ maps the point $x \in B_d$ to $T(x) \triangleq \frac{1}{\sqrt{2}}(x, \|x\|_2^2)$ where $(x, \|x\|_2^2)$ is the $(d+1)$-dimensional vector formed by appending $\|x\|_2$ to $x$. $Q$ maps the point $q \in B_d$ to $Q(q) \triangleq \frac{1}{\sqrt{5}}(2q, -1)$. We now have the following two claims.

**Lemma C.1.** *Let $X = \{x_1, x_2, \ldots, x_k\}$ be a set of points in $B_d$. Then for any $q \in B_d$, if $x^* = \arg\min_{x \in X} \|q - x\|_2$, it is also true that $x^* = \arg\max_{x \in X} \langle T(x), Q(q) \rangle$.*

*Proof.* Consider two points $x, x' \in B_d$. It suffices to show that if $\|q - x\| \leq \|q - x'\|$, then $\langle T(x), Q(q) \rangle \geq \langle T(x'), Q(q) \rangle$.

To see this, note that we can rewrite $\|q - x\|^2 \leq \|q - x'\|^2$ in the form $\langle q - x, q - x \rangle \leq \langle q - x', q - x' \rangle$, which we can in turn simplify to get

$$-2\langle q, x \rangle + \|x\|^2 \leq -2\langle q, x' \rangle + \|x'\|^2. \tag{1}$$

But the LHS of (1) is simply $-\sqrt{10}\langle T(x), Q(q) \rangle$ while the RHS of (1) is likewise $-\sqrt{10}\langle T(x'), Q(q) \rangle$. Equation (1) thus implies that $\langle T(x), Q(q) \rangle \geq \langle T(x'), Q(q) \rangle$, as desired. □

**Lemma C.2.** *Let $x$, $x'$, and $q$ be points in $B_d$. Let*

$$\ell_1 \triangleq |\|q - x\| - \|q - x'\||$$

*and*

$$\ell_2 \triangleq |\langle T(x), Q(q) \rangle - \langle T(x'), Q(q) \rangle|.$$

*Then $\ell_1 \leq 2\sqrt{\ell_2}$.*

*Proof.* Note that $\ell_1 \leq \|q - x\| - \|q - x'\|$, so in particular

$$\ell_1^2 \leq \left| \|q - x\|^2 - \|q - x'\|^2 \right|. \tag{2}$$

Via the same logic in the proof of Lemma C.1, we can rewrite the RHS of (2) as $\sqrt{10}\ell_2$. It follows that $\ell_1^2 \leq \sqrt{10}\ell_2$ and thus that $\ell_1 \leq 2\sqrt{\ell_2}$. □

With these two lemmas, we can prove the following reduction.

**Theorem 15.** *Let $\mathcal{A}$ be an algorithm for learning nearest-neighbor partitions under the similarity metric $\delta(x, y) = -\langle x, y \rangle$ with $\alpha = 1/2$ that achieves a total loss of at most $R(k, d)$. Then there exists an algorithm $\mathcal{A}'$ for learning nearest-neighbor partitions under the similarity metric $\delta(x, y) = \|x - y\|_2$ (and $\alpha = 1$) that achieves a total loss of at most $2R(k, d+1)$.*

*Proof.* To construct algorithm $\mathcal{A}'$ from algorithm $\mathcal{A}$, we simply map each incoming query $q_t \in B_d$ for algorithm $\mathcal{A}'$ to the point $q_t' = Q(q_t) \in B_{d+1}$ and feed it to $\mathcal{A}$ (returning the label that $\mathcal{A}$ outputs, and providing $\mathcal{A}$ with the true label that we receive).

To see why this works, note that if the hidden centers for $\mathcal{A}'$ are the points $x_1, x_2, \ldots, x_k \in B_d$, then by Lemma C.1, all feedback we provide $\mathcal{A}$ is consistent with the set of hidden centers $T(x_1), T(x_2), \ldots, T(x_k) \in B_{d+1}$. Moreover, by Lemma C.2, whenever algorithm $\mathcal{A}$ incurs loss $\ell$, our algorithm $\mathcal{A}'$ incurs loss at most $2\ell$. It follows that $\mathcal{A}'$ incurs loss at most $2R(k, d+1)$. □

# D   Mistake Bounds

In this section we provide mistake bounds for our algorithms for learning linear classifiers and learning nearest neighbor partitions. In both cases we will get (near) state-of-the-art guarantees for the mistake bound, despite our algorithms being designed for the absolute loss function.

We begin by discussing our algorithm for learning linear classifiers. As noted in the introduction, note that since this algorithm incurs total loss of at most $O(d \log d)$, then we make at most $O(d \log d / \gamma)$ mistakes in a setting with margin $\gamma$. In the following theorem, we see that we can improve this bound to $O(d \log 1/\gamma + d \log d)$ (matching the mistake bound of the best halving-based algorithms whenever $\gamma \leq 1/d$).

**Theorem 16.** *Assume every query point $q_t$ we are provided satisfies $|\langle q_t, x_1 - x_2 \rangle| \geq \gamma$ for some $\gamma > 0$. Then the algorithm of Theorem 1 makes at most $O(d \log 1/\gamma + d \log d)$ mistakes.*

*Proof.* The main observation is that since we query our contextual search subroutine (which is trying to learn the hidden point $w = x_1 - x_2$) with the point $q_t$, if $|\langle q_t, x_1 - x_2 \rangle| \geq \gamma$, then either 1. we already know for certain the sign of $\langle q_t, x_1 - x_2 \rangle$, or 2. the width $\mathsf{width}(K_t; q_t) \geq \gamma$.

In the first case, we cannot make a mistake. In the second case, since the width is at least $\gamma$, it suffices to only consider the $\lceil \log(1/\gamma) \rceil$th term of the potential function in Theorem 14. Specifically, note that by Lemma A.1, if we let $i = \lceil \log(1/\gamma) \rceil$ then we have that

$$\mathsf{Vol}(K_{t+1} + z_i B_d) \leq \frac{3}{4} \mathsf{Vol}(K_t + z_i B_d).$$

In particular, since $z_i = 2^{-i}/8d$, our total number of errors is at most

$$\log_{4/3} \frac{\mathsf{Vol}\,(B_d + z_i B_d)}{\mathsf{Vol}(z_i B_d)} = d \log_{4/3} \left(1 + \frac{1}{z_i}\right) = O(d \log(1/\gamma) + d \log d).$$

$\square$

The same logic extends to learning nearest-neighbor partitions via the reduction in Theorem 11.

**Theorem 17.** *Assume every query point $q_t$ satisfies $\delta(q_t, R_i) > \gamma$ for all $i$ such that $q_t \notin R_i$. Then the algorithm of Theorem 2 makes at most $O(k^2 d(\log 1/\gamma + \log d))$ mistakes.*

*Proof.* We apply the reduction of Theorem 11 to Theorem 16. In particular, note that the analysis of Theorem 16 implies that our original algorithm for learning linear classifiers can be thought of (under these margin conditions) as a potential-based algorithm with loss function $L(S_t, q_t) = 1$ and with potential function

$$\Phi(S_t) = \log_{4/3} \frac{\mathsf{Vol}\,((1 + z_i) B_d)}{\mathsf{Vol}(S_t + z_i B_d)}.$$

The analysis of Theorem 16 combined with the guarantees of Theorem 11 imply a mistake bound of $O(k^2 d(\log 1/\gamma + \log d))$.

$\square$

Finally, we prove a general reduction for the notion of robust mistake bound defined in the introduction. Formally, the robust mistake bound with margin $\gamma$ is the loss induced by the loss function $\ell'(q, R_i) = \mathbf{1}(\delta(q, R_i) - \delta(q, R^*) \geq \gamma)$ (where $R^*$ is the region containing $q$). In the below lemma, we relate this to the loss induced by our standard loss function $\ell(q, R_i) = (\delta(q, R_i) - \delta(q, R^*))$.

**Lemma D.1.** *If an algorithm has total loss at most $R$ under the loss function $\ell(q, R_i) = (\delta(q, R_i) - \delta(q, R^*))$, it has a robust mistake bound of $O(R/\gamma)$ under margin $\gamma$.*

*Proof.* This immediately follows form the fact that:

$$\ell(q, R_i) = (\delta(q, R_i) - \delta(q, R^*)) \geq \gamma \cdot \mathbf{1}(\delta(q, R_i) - \delta(q, R^*) \geq \gamma) = \gamma \ell'(q, R_i).$$

$\square$

# E   Omitted Proofs

## E.1   Omitted Proofs from Section 3.2

*Proof of Theorem 8.* The general idea behind this reduction is simple. We will run our algorithm $\mathcal{A}$ for contextual search to find the hidden point $w = x_1 - x_2$. Whenever we are given a query point $q_t$ and need to guess the sign of $\langle q_t, w \rangle$, we will ask our contextual search algorithm $\mathcal{A}$ for a guess $g_t$ for the value of $\langle q_t, w \rangle$. If $g_t > 0$, we will guess that $\langle q_t, w \rangle > 0$; otherwise, we will guess that $g_t < 0$.

There is one important caveat here: how do we update the algorithm $\mathcal{A}$? Recall that any contextual search algorithm expects binary feedback each round as to whether its guess $g_t$ was too high or too low. While in some cases, we can provide $\mathcal{A}$ with accurate feedback, in many cases we cannot: for example, if $\mathcal{A}$ submits a guess $g_t = 2.5$ for $\langle q_t, w \rangle$, and all we learn is that $\langle q_t, w \rangle \geq 0$, we cannot say with confidence whether $\mathcal{A}$'s guess was too large or not.

The solution to this is to *only update the state of $\mathcal{A}$ on rounds where we guess the sign incorrectly*. Note that for such rounds, we definitively know whether $g_t$ was too high or too low; for example, if $\mathcal{A}$ guessed $g_t = 2.5$ and hence we guess $\langle q_t, w \rangle > 0$, but it turns out that $\langle q_t, w \rangle < 0$, we know for certain that the guess $g_t$ was too high. On all other rounds we do not update the state of $\mathcal{A}$, effectively rolling back the state of $\mathcal{A}$ to before we asked the question about $q_t$. This means that the effective number of rounds $\mathcal{A}$ experiences (gets feedback on) may be less than $T$; nonetheless, since $R(d, T)$ is non-decreasing in $T$, the total loss of $\mathcal{A}$ on these rounds is still at most $R(d, T)$.

Finally, we will relate the loss of our algorithm $\mathcal{A}'$ for learning nearest neighbors to the loss of the contextual search algorithm $\mathcal{A}$. To start, note that we only sustain loss in rounds when we guess the sign of $\langle q_t, w \rangle$ incorrectly. Luckily, these rounds happen to be exactly the rounds where we update the state of $\mathcal{A}$ (and thus the rounds whose loss counts towards the $R(d, T)$ bound). In a round where we guess the sign incorrectly, $\mathcal{A}'$ sustains a loss of $|\langle q_t, w \rangle|^\alpha$, and $\mathcal{A}$ sustains a loss of $|\langle q_t, w \rangle - g_t|^\alpha$. Since $\mathsf{sign}(g_t) \neq \mathsf{sign}(\langle q_t, w \rangle)$, this means that $|\langle q_t, w \rangle - g_t| \geq |\langle q_t, w \rangle|$, and therefore $\mathcal{A}$ sustains more loss than $\mathcal{A}'$. It follows that the total loss sustained by $\mathcal{A}'$ is at most the total loss sustained by $\mathcal{A}$ on this set of rounds, which in turn is at most $R(d, T)$. $\quad\square$

*Proof of Lemma 3.2.* To show the existence of such a distribution $v$, we will show that the following linear program has a solution.

$$\min 0$$
$$Mv \geq 0$$
$$\sum_i v_i = 1$$
$$v \geq 0$$

If the above program has no solution then by the Strong Duality Theorem (see [MG07] section 6.1), we know that the dual program below is unbounded. (Since the dual has a feasible solution $y = 0, z = 0$, we know that it must be unbounded). In particular, for any value $z > 0$, there exists a corresponding solution to $y$ to the dual program.

$$\max z$$
$$M^T y + z\mathbf{1} \leq 0$$
$$y \geq 0$$

Let $y$ be any solution to the above dual with $z = 1$. Since $M + M^T$ has all non-negative entries, we know that $(M + M^T)y \geq 0$. Combining this with the fact that $M^T y \leq -\mathbf{1}$, we get that $My \geq 0$. This contradicts the fact that there is no solution to the primal since $\frac{y}{\sum_i y_i}$ is a feasible point. $\quad\square$

## E.2   Omitted Proofs from Section 4.1

Here we prove Theorem 12. Fix an even integer $p$. We define the following kernel maps.

**Definition 18.** *Let* $\mathsf{Ker} : \mathbb{R} \to \mathbb{R}^{p+1}$ *be the map defined by*

$$\mathsf{Ker}(x) \triangleq (1, x, x^2, \ldots, x^p).$$

**Definition 19.** *For a point* $y = (y_1, \ldots, y_d) \in B_d$, *define the map* $G : B_d \to B_{(p+1)d}$ *as*

$$G(y) \triangleq \frac{1}{\sqrt{pd}}\left(\mathsf{Ker}\left(\frac{y_1}{p}\right), \ldots, \mathsf{Ker}\left(\frac{y_d}{p}\right)\right)$$

*where above the outputs of* $\mathsf{Ker}(\cdot)$ *are simply concatenated.*

**Definition 20.** *Let* $F : \mathbb{R} \to \mathbb{R}^{p+1}$ *be defined by*

$$F(a) \triangleq \left(a^p, -\binom{p}{1}a^{p-1}, \ldots, -\binom{p}{p-1}a, \binom{p}{p}\right)$$

Note that if $|a| \leq 1/p$, all components of $F(a)$ are at most 1 in absolute value.

**Definition 21.** *For a point $z = (z_1, \ldots, z_d) \in \mathbb{R}^d$, define the map $H : B_d \to B_{(p+1)d}$ as*

$$H(z) \triangleq \frac{1}{\sqrt{pd}} \left( F\left(\frac{z_1}{p}\right), \ldots, F\left(\frac{z_d}{p}\right) \right).$$

The key property that these maps satisfy is stated below.

**Lemma E.1.** *For points $y, z \in \mathbb{R}^d$,*

$$\langle G(y), H(z) \rangle = \frac{1}{pd} \left\| \frac{y-z}{p} \right\|_p^p.$$

*Proof.* The proof follows by substituting in the definitions for $G$ and $H$ and using the binomial theorem. □

In particular, we can rewrite the statement of Lemma E.1 in the form

$$\|y - z\|_p = p^{(p+1)/p} d^{1/p} \langle G(y), H(z) \rangle^{1/p}. \tag{3}$$

This suggests a reduction to the inner-product similarity metric similar to the reduction for Euclidean norm in Theorem 15. In particular, note that for $p \geq 1$, we have that:

$$|x|^p - |y|^p \geq |x - y|^p.$$

In particular, this implies that

$$||q-x|| - ||q-x^*|| \leq (||q-x||^p - ||q-x^*||^p)^{1/p} = p^{(p+1)/p} d^{1/p} |\langle G(q), H(x^*) \rangle - \langle G(q), H(x) \rangle|^{1/p}.$$

Thus, if we use our map $H$ to map each query point $q_t$ to the point $H(q_t) \in B_{p(d+1)}$ and feed it into an algorithm with similarity metric $\delta(x, y) = -\langle x, y \rangle$ and $\alpha = 1/p$, Lemma E.1 implies that this algorithm will successfully learn the partition induced by the points $G(x_i)$. We can now complete the proof of Theorem 12.

*Proof of Theorem 12.* From Corollary 3.3, there exists an algorithm for learning nearest neighbor partitions with similarity function $\delta(x, y) = -\langle x, y \rangle$ and $\alpha = 1/p$ with expected total loss $O(p^2 k^2 d \log d)$. Applying this algorithm as described above (noting that the ambient dimension is now $p(d+1)$ and the loss is scaled by a factor of $O(pd^{1/p})$), we obtain the bound in the statement. □

### E.3 Omitted Proofs from Section 4.2

Here, we prove Theorem 13.

#### E.3.1 Kernelization

Fix the $L^p$ norm that we are working with. Let $p' = \lfloor p \rfloor + 1$. For each $i = 1, 2, \ldots$, let

$$\delta_i = \frac{1}{100d^2 p' 2^i}, D_i = \frac{1}{2\delta_i}.$$

We will now define two maps that will be crucial for our algorithm.

For each $i$, we will define two maps $G_i, H_i$ that map points in $\mathbb{R}^d$ to points in $\mathbb{R}^{p'd(2D_i+1)}$ such that for two points $y, z \in \mathbb{R}^d$, the images $G_i(y), H_i(z)$ satisfy the property that $\langle G_i(y), H_i(z) \rangle$ is a good approximation of $\|y - z\|_p^p$. We need to consider maps for different values of $i$ because as $i$ increases, the approximation gets better but the dimension of the image space also increases. We can think of these maps for different values of $i$ as approximations at different scales.

Through the next several definitions, we build the first map $G_i$.

**Definition 22.** *For $x \neq 0$, let $\mathsf{sign}(x) \triangleq x/|x|$. Let $\mathsf{sign}(0) \triangleq 0$.*

**Definition 23.** *Let $D : \mathbb{R} \to \mathbb{R}^{p'}$ be the map defined by*

$$D(x) = (|x|^p, \mathsf{sign}(x)\,|x|^{p-1}, |x|^{p-2}, \mathsf{sign}(x)\,|x|^{p-3}, \dots, \mathsf{sign}(x)^{\lfloor p \rfloor}\,|x|^{p-\lfloor p \rfloor})\,.$$

**Definition 24.** *Let $\mathsf{Ker}_i : \mathbb{R} \to \mathbb{R}^{p'(2D_i+1)}$ be the map defined by*

$$\mathsf{Ker}_i(x) = (D(x+0.5), D(x+0.5-\delta_i), \dots, D(x-0.5))$$

*where the tuples given by the output of $D(\cdot)$ are simply concatenated.*

**Definition 25.** *For a point $y = (y_1, \dots, y_d) \in B_d$, define the map $G_i : \mathbb{R}^d \to \mathbb{R}^{p'd(2D_i+1)}$ as*

$$G_i(y) = (\mathsf{Ker}_i(y_1/2), \dots, \mathsf{Ker}_i(y_d/2))\,.$$

Now we build the second map $H_i$ through the next set of definitions.

**Definition 26.** *Let $F_i : [-1/2, 1/2] \to \mathbb{R}^{p'(2D_i+1)}$ be the map defined as follows. Assume that we want to compute $F_i(x)$. Then perform the following steps.*

- *Let $c$ be the unique integer such that $c\delta_i \le x < (c+1)\delta_i$*
- *For any element of $\mathbb{R}^{p'(2D_i+1)}$, group the coordinates into consecutive groups of $p'$ and label the groups with $-D_i, -D_i+1, \dots, D_i$*
- *Let $F_i(x)$ be the element of $\mathbb{R}^{p'(2D_i+1)}$ where*
  - *The group labeled $c$ is set to*

$$\left(1, p(x-c\delta_i), \frac{p(p-1)}{2}(x-c\delta_i)^2, \frac{p(p-1)(p-2)}{6}(x-c\delta_i)^3, \dots, \right)$$

  - *All other groups are set to $(0, 0, \dots, 0)$*

**Definition 27.** *For a point $z = (z_1, \dots, z_d) \in B_d$, define the map $H_i : \mathbb{R}^d \to \mathbb{R}^{p'd(2D_i+1)}$ as*

$$H_i(z) = (F_i(z_1/2), \dots, F_i(z_d/2))\,.$$

The intuition for the interplay between the maps $G_i$ and $H_i$ is that the "kernel" map $G_i$ discretizes the function $|x|^p$ as well as its derivatives and then $H_i$ takes the first $p'$ terms of the Taylor series expansion at the closest point in the discretization.

Formally, the key property that the maps $G_i, H_i$ satisfy is the following:

**Lemma E.2.** *For points $y, z \in B_d$,*

$$\left| \langle G_i(y), H_i(z) \rangle - \left\| \frac{y-z}{2} \right\|_p^p \right| \le d(p\delta_i)^p$$

The proof of Lemma E.2 relies on the following inequality.

**Claim 28.** *Let $p > 2$. Then for any $x, x' \in [-1, 1]$, we have the inequality*

$$\left| |x|^p - \sum_{i=0}^{\lfloor p \rfloor} \frac{p(p-1)\dots(p-i+1)}{i!}(x-x')^i \mathsf{sign}(x')^i\,|x'|^{p-i} \right| \le (p|x-x'|)^p\,.$$

*Proof.* Note that the function $|x|^p$ is $\lfloor p \rfloor$-times continuously differentiable and its derivatives are are $p \cdot \mathsf{sign}(x)\,|x|^{p-1}, p(p-1)\,|x|^{p-2}, \dots$ and so on. Thus, we may write

$$|x|^p = |x'|^p + \int_{x'}^{x} p \cdot \mathsf{sign}(y)\,|y|^{p-1}\,dy$$

$$= |x'|^p + p(x-x')\mathsf{sign}(x')\,|x'|^{p-1} + \int_{x'}^{x}\int_{x'}^{y_1} p(p-1)\,|y_2|^{p-2}\,dy_2 dy_1$$

$$\vdots$$

$$= \sum_{i=0}^{\lfloor p \rfloor} \frac{p(p-1)\dots(p-i+1)}{i!}(x-x')^i \mathsf{sign}(x')^i\,|x'|^{p-i}$$

$$+ \int_{x'}^{x} \cdots \int_{x'}^{y_{\lfloor p \rfloor-1}} p \cdots (p-\lfloor p \rfloor+1)(\mathsf{sign}(y_{\lfloor p \rfloor})^{\lfloor p \rfloor}\,|y_{\lfloor p \rfloor}|^{p-\lfloor p \rfloor} - \mathsf{sign}(x')^{\lfloor p \rfloor}\,|x'|^{p-\lfloor p \rfloor})dy_{\lfloor p \rfloor} \dots dy_1\,.$$

It now suffices to bound the last term which is the "error" term.. However since $p - \lfloor p \rfloor < 1$,

$$\left| \mathsf{sign}(y)^{\lfloor p \rfloor} |y|^{p - \lfloor p \rfloor} - \mathsf{sign}(x')^{\lfloor p \rfloor} |x'|^{p - \lfloor p \rfloor} \right| \leq |y - x'|^{p - \lfloor p \rfloor} \ ,$$

so the error term is at most

$$p \cdots (p - \lfloor p \rfloor + 1)|x - x'|^p \leq (p|x - x'|)^p \ ,$$

and now we immediately get the desired inequality. $\qquad \square$

Now we can prove Lemma E.2.

*Proof of Lemma E.2.* Let $y = (y_1, \ldots, y_d)$ and $z = (z_1, \ldots, z_d)$. For each $j \in [d]$ let $c_j$ be the integer such that $c_j \delta_j \leq 0.5 z_j < (c_j + 1) \delta_i$. Now we have

$$\langle G_i(y), H_i(z) \rangle = \sum_{j \in [d]} \left( \sum_{i=0}^{\lfloor p \rfloor} \frac{p(p-1) \ldots (p - i + 1)}{i!} (0.5 z_j - c_j \delta_i)^i \mathsf{sign}(0.5 y_j - c_j \delta_i)^i |0.5 y_j - c_j \delta_i|^{p-i} \right).$$

Now we can apply Claim 28 with $x = 0.5(y_j - z_j)$ and $x' = 0.5 y_j - c_j \delta_i$ to bound each term. Note that $|x - x'| < \delta_i$. Since the sum contains $d$ terms, we immediately get the desired conclusion. $\quad \square$

Our full algorithm for learning nearest neighbor partitions in $L^p$ norm is described below.

### E.3.2 Algorithm

---
**Algorithm 2** MULTISCALE NEAREST NEIGHBOR LEARNING FOR $L^p$ NORMS
---
There are two unknown points $x_1, x_2 \in B_d$
For each $i = 1, 2, \ldots$ initialize the sets $S_i = [-1, 1]^{2p'd(2D_i + 1)}$. Note that

$$S_i \supset \{(G_i(x), G_i(y)) | x, y \in [-1/2, 1/2]^d\}$$

**for** $t$ in $1, 2, \ldots, T$ **do**
    Adversary picks $q_t \in B_d$
    **for** $i = 1, 2, \ldots$ **do**
        Let $v_{i,t} = (-H_i(q_t), H_i(q_t))$
        Let $w_{i,t}$ be the width of the set $S_i$ in direction $v_{i,t}$ i.e.

$$w_{i,t} = \max_{u \in S_i} \langle v_{i,t}, u \rangle - \min_{u \in S_i} \langle v_{i,t}, u \rangle$$

    Let $i_t$ be the smallest integer $i$ such that $w_{i,t} \geq 10^3 D_i p d^2 (p \delta_i)^p$
**if** $\mathsf{Vol}\left(\{z \in S_{i_t}, v_{i_t,t} \cdot z > 0\}\right) \geq \mathsf{Vol}\left(\{z \in S_{i_t}, v_{i_t,t} \cdot z < 0\}\right)$ **then**
    Guess label 1
**if** $\mathsf{Vol}\left(\{z \in S_{i_t}, v_{i_t,t} \cdot z > 0\}\right) < \mathsf{Vol}\left(\{z \in S_{i_t}, v_{i_t,t} \cdot z < 0\}\right)$ **then**
    Guess label 2
**if** true label is 1 **then** update

$$S_{i_t} \leftarrow \{z \in S_{i_t}, v_{i_t,t} \cdot z > -3d(p \delta_{i_t})^p\}$$

**if** true label is 2 **then** update

$$S_{i_t} \leftarrow \{z \in S_{i_t}, v_{i_t,t} \cdot z < 3d(p \delta_{i_t})^p\}$$

---

**Remark E.3.** *In the algorithm, it is stated that we keep track of sets $S_i$ for all integers $i$. Technically, this is not possible as there are infinitely many sets to keep track of but it will be clear from the analysis that it suffices to track the set $S_i$ only for $i \leq \mathrm{poly}(p, d, T)$ and if $i_t$ is too large, we can simply guess arbitrarily and our loss will be upper bounded by $1/T$.*

As in the previous section, for each timestep $t$ and integer $i$, we let $S_i^{(t)}$ denote the set $S_i$ at the beginning of timestep $t$ in the execution of the algorithm.

**Claim 29.** *For all $i$ and all timesteps $t$, the set $S_i^{(t)}$ contains the $L^2$ ball of radius $0.1(p\delta_i)^p$ centered around $(G_i(x_1), G_i(x_2))$ where $x_1, x_2$ are the two unknown centers.*

*Proof.* We will prove the claim by induction on $t$. The base case is obvious. Now we do the induction step. Consider a timestep $t$. Assume that the adversary gives us the point $q_t$. WLOG the true label of $q_t$ is 1 i.e.

$$\|q_t - x_1\|_p^p \le \|q_t - x_2\|_p^p \; .$$

Note that $\|v_{i,t}\|_2 \le 10d$ for all $i, t$ (this uses the fact that in Definition 26, $|x - c\delta_i| \le \delta_i \le 1/p$). Thus, any point $z$ in the ball of radius $0.1(p\delta_i)^p$ centered around $(G_i(A), G_i(B))$ satisfies

$$|v_{i,t} \cdot z - v_{i,t} \cdot (G_i(x_1), G_i(x_2))| \le d(p\delta_i)^p \; .$$

However

$$v_{i,t} \cdot (G_i(x_1), G_i(x_2)) = - \langle H_i(q_t), G_i(x_1) \rangle + \langle H_i(q_t), G_i(x_2) \rangle$$

and we can now use Lemma E.2 to deduce

$$\left| \left( - \left\| \frac{q_t - x_1}{2} \right\|_p^p + \left\| \frac{q_t - x_2}{2} \right\|_p^p \right) - v_{i,t} \cdot (G_i(x_1), G_i(x_2)) \right| \le 2d(p\delta_i)^p \; .$$

Thus, by the triangle inequality, we must actually have for all $z$ in the ball of radius $0.1(p\delta_i)^p$ centered around $(G_i(x_1), G_i(x_2))$,

$$v_{i,t} \cdot z \ge -3d(p\delta_i)^p$$

which implies that all of these points are all contained in $S_i$ after the update step, completing the induction. $\square$

We will also need the following geometric fact.

**Lemma E.4** (From [LLS20]). *Let $S \subset \mathbb{R}^d$ be a convex polytope and $v$ be a unit vector. Assume that the width of $S$ in direction $v$ is at least $8d\epsilon$. Then any strip of width $\epsilon$ normal to direction $v$ contains at most $1/4$ of the volume of $S$.*

*Proof.* Let $C$ be a cross section of $S$ normal to direction $v$ with maximal area. Let $u_1, u_2$ be two points in $S$ that minimize and maximize the inner product with $v$ respectively. Then either $u_1$ or $u_2$ is distance at least $4d\epsilon$ from the hyperplane containing $C$. WLOG $u_1$ is at least $4d\epsilon$ away from this hyperplane. Then the cone containing $u_1$ and $C$ must be contained in $S$ (since $S$ is convex) so

$$\mathsf{Vol}(S) \ge 4\epsilon d \cdot \mathsf{Vol}(C) \cdot \frac{1}{d} = 4\epsilon \mathsf{Vol} C \; .$$

On the other hand, by the maximality of $C$, the volume contained in any $\epsilon$-width strip is at most $\epsilon \mathsf{Vol}(C)$ so we are done. $\square$

**Claim 30.** *Consider a timestep $t$. If the learner incurs nonzero loss then*

$$\mathsf{Vol}\left( S_{i_t}^{(t+1)} \right) \le \frac{3}{4} \mathsf{Vol}\left( S_{i_t}^{(t)} \right) \; .$$

*Proof.* Without loss of generality, the true label is 1 and our guess was 2. Then we must have

$$\mathsf{Vol}\left( \{z \in S_{i_t}^{(t)}, v_{i_t,t} \cdot z > 0\} \right) \le \frac{1}{2} \mathsf{Vol}\left( S_{i_t}^{(t)} \right) \; .$$

Also since $w_{i,t} \ge 10^3 D_{i_t} pd^2 (p\delta_{i_t})^p$ and the dimension of the space in which $S_{i_t}^{(t)}$ lives is $2p'd(2D_{i_t} + 1)$, Lemma E.4 gives us that

$$\mathsf{Vol}\left( \{z \in S_{i_t}^{(t)}, -3d(p\delta_{i_t})^p < v_{i_t,t} \cdot z < 0\} \right) \le \frac{1}{4} \mathsf{Vol}\left( S_{i_t}^{(t)} \right) \; .$$

Thus, we deduce that

$$\mathsf{Vol}(S_{i_t}^{(t+1)}) = \mathsf{Vol}\left( \{z \in S_{i_t}^{(t)}, v_{i_t,t} \cdot z > -3d(p\delta_{i_t})^p\} \right) \le \frac{3}{4} \mathsf{Vol}\left( S_{i_t}^{(t)} \right) \; ,$$

as desired. $\square$

**Theorem 31.** *The total loss incurred by* MULTISCALE NEAREST NEIGHBOR LEARNING *is at most*

$$\frac{\text{poly}(d,p)}{\|x_1 - x_2\|_p} \cdot \left(\frac{1}{p-2}\right)^2 .$$

*Proof.* Combining Claim 29 and Claim 30 implies that for any index $i$, the number of times that $i_t = i$ and we incur nonzero loss is at most

$$O\left(\log\left(\frac{2^{2p'd(2D_i+1)}}{(0.1(p\delta_i)^p)^{2p'd(2D_i+1)}\,\text{Vol}(B_{2p'(2D_i+1)}(0,1))}\right)\right) = O\left(p^3 d^3 2^i (i + \log dp)\right) .$$

Note that if $i_t = i$, then by the definition of our algorithm

$$w_{i-1,t} = \max_{u \in S_{i-1}^t} \langle v_{i-1,t}, u\rangle - \min_{u \in S_{i-1}^t} \langle v_{i-1,t}, u\rangle \le 10^3 D_{i-1} p d^2 (p\delta_{i-1})^p .$$

Note that the origin is clearly always contained in $S_{i-1}$ so using Claim 29, we get that

$$|\langle v_{i-1,t}, (G_{i-1}(x_1), G_{i-1}(x_2))\rangle| \le 10^3 D_{i-1} p d^2 (p\delta_{i-1})^p = O(d^2 p^2 (p\delta_{i-1})^{p-1}) .$$

Lemma E.2 implies that

$$\left|\left(-\left\|\frac{q_t - x_1}{2}\right\|_p^p + \left\|\frac{q_t - x_2}{2}\right\|_p^p\right) - v_{i-1,t} \cdot (G_{i-1}(x_1), G_{i-1}(x_2))\right| \le 2d(p\delta_{i-1})^p ,$$

so we deduce that

$$\left|\|q_t - x_1\|_p^p - \|q_t - x_2\|_p^p\right| \le O(d^2 p^2 (4p\delta_{i-1})^{p-1}) \le O(d^2 p^2 (10 \cdot 2^i d^2)^{-(p-1)}) .$$

Note that our loss at each round may be bounded as

$$\left|\|q_t - x_1\|_p - \|q_t - x_2\|_p\right| \le \frac{\left|\|q_t - x_1\|_p^p - \|q_t - x_2\|_p^p\right|}{\max(\|q_t - x_1\|_p, \|q_t - x_2\|_p)^{p-1}} \le \frac{\left|\|q_t - x_1\|_p^p - \|q_t - x_2\|_p^p\right|}{\|0.5(x_1 - x_2)\|_p^{p-1}} .$$

Alternatively, we may also use the trivial bound

$$\left|\|q_t - x_1\|_p - \|q_t - x_2\|_p\right| \le \|x_1 - x_2\|_p .$$

Let $i_0$ be the largest positive integer such that

$$\frac{i_0}{2^{i_0}} \ge 0.1 d^{-1} \|x_1 - x_2\|_p .$$

Note that $i_0 \ge 2$.

We can now bound the total loss of our algorithm, say $L$, as follows:

$$L \le \sum_{i=1}^{i_0} \|x_1 - x_2\|_p\, O\left(p^3 d^3 2^i (i + \log dp)\right)$$

$$+ \sum_{i=i_0+1}^{\infty} \frac{\left|\|q_t - x_1\|_p^p - \|q_t - x_2\|_p^p\right|}{\|0.5(x_1 - x_2)\|_p^{p-1}} \cdot O\left(p^3 d^3 2^i (i + \log dp)\right)$$

$$\le \sum_{i=1}^{i_0} \|x_1 - x_2\|_p\, O\left(p^3 d^3 2^i (i + \log dp)\right)$$

$$+ \frac{1}{\|0.5(x_1 - x_2)\|_p^{p-1}} \sum_{i=i_0+1}^{\infty} O(d^2 p^2 (10 \cdot 2^i d^2)^{-(p-1)}) \cdot O\left(p^3 d^3 2^i (i + \log dp)\right)$$

$$\le \text{poly}(d,p) \frac{1}{\|x_1 - x_2\|_p} \sum_{i=0}^{\infty} \frac{1+i}{2^{(p-2)i}}$$

$$\le \text{poly}(d,p) \cdot \frac{1}{\|x_1 - x_2\|_p} \cdot \frac{1}{(1 - 2^{-(p-2)})^2}$$

$$\le \frac{\text{poly}(d,p)}{\|x_1 - x_2\|_p} \cdot \left(\frac{1}{p-2}\right)^2 .$$

$\square$

*Proof of Theorem 13.* In light of Theorem 11, it suffices to argue that MULTISCALE NEAREST NEIGHBOR LEARNING is a potential-based algorithm. Indeed, the corresponding potential is defined as follows. Let $i_0$ be the largest positive integer such that

$$\frac{i_0}{2^{i_0}} \geq 0.1 d^{-1} \Delta \,.$$

Define

$$P_t = \sum_{i=1}^{i_0} \Delta \log \frac{\mathsf{Vol}\left(S_i^{(t)}\right)}{\mathsf{Vol}(B_{2p'(2D_i+1)}(0, 0.1 \cdot (p\delta_i)^p))}$$

$$+ \sum_{i=i_0+1}^{\infty} \frac{d^2 p^2}{(10 \cdot 2^i d^2)^{p-1}(0.5\Delta)^{p-1}} \log \frac{\mathsf{Vol}\left(S_i^{(t)}\right)}{\mathsf{Vol}(B_{2p'(2D_i+1)}(0, 0.1 \cdot (p\delta_i)^p))} \,.$$

The proof of Theorem 31 immediately implies that $P_t$ is a valid potential and we are done. □

# F    Learning General Convex Regions: Missing Proofs

In this appendix, we present the full proof of Theorem 4.

Our construction is based on only picking points on the surface of the unit ball (i.e. the unit hypersphere). There are two key factors to ensuring that the algorithm accrues enough total error; we need to ensure that (i) each time we choose a point, it could lie in either of the two regions and (ii) the point is sufficiently far from the region it is not in. To guarantee the former, we choose our points by considering a separating hyperplane between the two regions so far.

To guarantee the latter, we will choose our points to be $\epsilon$-far from each other, for some $\epsilon$ based on the total number of points we need to choose $T$. This, combined with the fact that we chose points on the surface of the unit ball, implies that the minimum penalty for a mistake is $\Omega(\epsilon^2)$. Note that for both of our guarantees to simultaneously work out, we actually need our separating hyperplane to go through the origin (so the resulting intersection has enough surface area for this part of the argument).

The requirement that points be $\epsilon$ far from each other limits the total number of points we can choose. Roughly speaking, each point removes on the order of a $\epsilon^{d-2}$-fraction of the (hyper-)surface area of the intersection of the separating hyperplane with the unit hypersphere. Maximizing $\epsilon$ while ensuring we can pick $T$ points yields the desired bound in the theorem statement.

Our construction will utilize two technical results concerning the geometry of high-dimensional objects. One is a result of Klee regarding the existence of separating hyperplanes for convex cones [Kle55]. It uses the following notation.

**Definition 32** ([Kle55]). *A 0-cone is a closed convex cone having the origin (denoted 0) as its vertex. For a 0-cone $A$, $A'$ denotes the linear subspace $A \cap -A$.*

The technical lemma gives conditions for a strict linear separator between two such convex cones.

**Theorem 33** ([Kle55] Theorem 2.7). *Suppose $E$ is a separable normed linear space, $A$ and $B$ are 0-cones in $E$, $A$ is locally compact, and $A \cap B = \{0\}$. Then $E$ admits a continuous linear functional $\mathcal{H}$ such that $\mathcal{H} < 0$ on $A \setminus A'$, $\mathcal{H} = 0$ on $A' \cup B'$, and $\mathcal{H} > 0$ on $B \setminus B'$.*

The next technical lemma upper bounds the surface area of a hypersphere cap relative to the entire hypersphere, implying that many such caps are needed to cover the hypersphere.

**Definition 34.** *We denote the $d$-dimensional unit hypersphere as $\mathcal{S}_d \triangleq \{x \in \mathbb{R}^d \mid \|x\|_2 = 1\}$ (note that this is the surface of the unit ball). We denote its (hyper-)surface area as $\mathcal{A}_d$.*

*Next, we denote the cap centered at $v \in \mathcal{S}_d$ of angle $\phi \in [0, \pi/2]$ as $\mathcal{S}_d(v, \phi) \triangleq \mathcal{S}_d \cap \{x \in \mathbb{R}^d \mid \langle x, v \rangle \leq \cos \phi\}$. We denote its (hyper-)surface area as $\mathcal{A}_d(\phi)$.*

**Lemma F.1.** *For all integer $d \geq 1$ and $\phi \in [0, \pi/2]$,*

$$\frac{\mathcal{A}_d(\phi)}{\mathcal{A}_d} \leq \phi^{d-1}$$

*Proof.* We will prove this bound by building on the analysis of [Li11]. The surface area of a $d$-dimensional hypersphere is well known to be

$$\mathcal{A}_d = \frac{2\pi^{d/2}}{\Gamma(d/2)},$$

where the gamma function $\Gamma$ represents the standard extension of the factorial function.

We begin with the following observation of Li:

$$\mathcal{A}_d(\phi) = \frac{2\pi^{(d-1)/2}}{\Gamma((d-1)/2)} \int_0^\phi \sin^{d-2} x \, dx.$$

Next, we apply the fact that $\sin x \leq x$ for $x \geq 0$.

$$\begin{aligned}
\mathcal{A}_d(\phi) &\leq \frac{2\pi^{(d-1)/2}}{\Gamma((d-1)/2)} \int_0^\phi x^{d-2} \, dx \\
&= \frac{2\pi^{(d-1)/2}}{\Gamma((d-1)/2)} \left[ \frac{1}{d-1} \phi^{d-1} \right] \\
&= \frac{\pi^{(d-1)/2}}{\Gamma((d+1)/2)} \phi^{d-1} \frac{\mathcal{A}_d(\phi)}{\mathcal{A}_d} \\
&\leq \frac{\Gamma(d/2)}{2\sqrt{\pi}\Gamma((d+1)/2)} \phi^{d-1}
\end{aligned}$$

We are almost done; we just need to show that $\frac{\Gamma(d/2)}{2\sqrt{\pi}\Gamma((d+1)/2)} \leq 1$. After $\Gamma(3/2)$, subsequent half-values of $\Gamma$ are increasing and so the desired claim is trivially true for $d \geq 3$. We manually check $d = 1$ and $d = 2$, noting that $\Gamma(1/2) = \sqrt{\pi}$, $\Gamma(1) = 1$, and $\Gamma(3/2) = \sqrt{\pi}/2$.

$$\begin{aligned}
\frac{\Gamma(1/2)}{2\sqrt{\pi}\Gamma(2/2)} &= \frac{1}{2} \\
\frac{\Gamma(2/2)}{2\sqrt{\pi}\Gamma(3/2)} &= \frac{1}{\pi}
\end{aligned}$$

This completes the proof. $\qquad\square$

We are now ready to give the full proof for Theorem 4, which is restated below for convenience.

**Theorem 4.** *Any algorithm for learning general convex regions incurs a total loss of at least* $\Omega\left(T^{(d-4)/(d-2)}\right)$, *even when there are only two regions.*

*Proof.* Our counterexample depends on two parameters: the dimension $d$ and an error parameter $\epsilon > 0$. We will choose $\epsilon = 1/T^{1/(d-2)}$, where $T$ is the total number of time steps.

Our construction restricts itself to choosing points on $\mathcal{S}_d$. The construction begins with the opposite points $x_1 = e_1 \triangleq (1, 0, 0, \ldots, 0)$ and $x_2 = -e_1$. $x_1$ has true label 1 and $x_2$ has true label 2. We will use $A$ to denote the conic hull of the points with true label 1 and $B$ to denote the conic hull of the points with true label 2. By construction, $A$ and $B$ are 0-cones. We will maintain the two invariants that (i) $A \cap B = \{0\}$ and (ii) $A' = B' = \{0\}$, which we will use when invoking Theorem 33.

We now explain how to generate subsequent points $x_t$ for $t \geq 3$. By Theorem 33, we know there is a separating hyperplane $\mathcal{H}$ that passes through the origin and strictly separates $A \setminus \{0\}$ from $B \setminus \{0\}$. We pick an arbitrary hyperplane $\mathcal{H}$ that satisfies the previous statement and examine its intersection with the hypersphere, $X \triangleq \{x \mid \mathcal{H}(x) = 0\} \cap \mathcal{S}_d$.

Since we $\mathcal{H}$ passes through the origin, $X$ is a $(d-1)$-dimensional unit hypersphere. We want to choose our next point $x_t$ to be a point in $X$ that is at least $\epsilon$-far from all previously chosen points $x_1, x_2, \ldots, x_{t-1}$. This condition rules out $(t-1)$ hypersphere caps of angles at most $2\arcsin\frac{\epsilon}{2}$. Observe that as $T$ increases and $\epsilon$ decreases, this angle bound scales as $O(\epsilon)$. By Lemma F.1, we have (hyper-)surface area remaining as long as $(t-1) \cdot O(\epsilon^{d-2}) \leq 1$, i.e. we can pick up to $(1/\epsilon)^{d-2} = T$ total points safely. We will pick an arbitrary point $x_t$ satisfying our $\epsilon$-far rule.

We assign this point $x_t$ a uniform random true label between $1$ and $2$. It remains to establish that our two invariants are still satisfied. For the sake of contradiction, assume that (i) is no longer true and that there is a non-origin point $z$ in the intersection $A \cap B$. Without loss of generality, $x_t$ was assigned true label 1 and $f$ used to be strictly positive (negative respectively) on $A \setminus \{0\}$ ($B \setminus \{0\}$ respectively). We can deduce the following.

$$\mathcal{H}(z) < 0$$

$$\mathcal{H}(z) = \mathcal{H}\left(\sum_{x \text{ has true label } 1} \alpha_x x\right)$$

$$= \sum_{x \text{ has true label } 1} \alpha_x \mathcal{H}(x)$$

$$\geq 0$$

for some values $\alpha_x \geq 0$. The first inequality follows from the fact that $z$ is in $B \setminus \{0\}$ and the second inequality follows from the fact that all points with true label 1 either are in $A \setminus \{0\}$ before this round or are $x_t$. We've reached a contradiction and conclude that invariant (i) remains true.

We establish invariant (ii) much in the same way. Again, for the sake of contradiction we assume that (ii) is false. Without loss of generality, assume that $x_t$ was assigned true label 1 and that now there exists a non-origin point $z$ in $A' = A \cap -A$. We follow a similar line of deduction.

$$\mathcal{H}(z) = \mathcal{H}\left(\sum_{x \text{ has true label } 1} \alpha_x x\right)$$

$$\mathcal{H}(z) \geq 0$$

$$\mathcal{H}(z) = \mathcal{H}\left(\sum_{x \text{ has true label } 1} \beta_x x\right)$$

$$\mathcal{H}(z) \leq 0$$

$$\mathcal{H}(z) = 0$$

for some values $\alpha_x \geq 0$ and $\beta_x \leq 0$. But this implies that $\alpha_x = \beta_x = 0$ forall $x$ that have true label 1 and are not the most recent point $x_t$. We are left with $z = \alpha_{x_t} x_t = \beta_{x_t} x_t$ for some $\alpha_{x_t} \geq 0$ and $\beta_{x_t} \leq 0$, but this can only hold if $\alpha_{x_t} = \beta x_t = 0$ (recall that $x_t$ is not the origin by construction). But then $z$ is the origin, which contradicts our assumption. Hence our assumption is false, i.e. invariant (ii) is indeed maintained.

All that remains is the add up error of our algorithm. Since we chose the true label for the final $T - 2$ points uniformly at random, any online algorithm will make $\frac{T-2}{2}$ mistakes in expectation.

How much error does the algorithm accrue for every mistake? Recall that our construction kept all points at least $\epsilon$ away from each other and all points were on the hypersphere (the surface of the unit ball). Figure 1 illustrates the situation with respect to one point. We can solve for the distance that our point of interest $x$ is away from the unit ball minus its cap, which is an upper bound for the convex hull of all other points.

$$h(2 - h) = w^2$$
$$2h = w^2 + h^2$$
$$2h = \epsilon^2$$
$$h = \epsilon^2/2$$

Hence the algorithm accrues $\Omega(\epsilon^2)$ expected error in each timestep, and so overall any online algorithm is expected to accrue $\Omega\left(T\epsilon^2\right)$ total error. By our choice of $\epsilon$, this equals the desired bound. $\qquad\square$

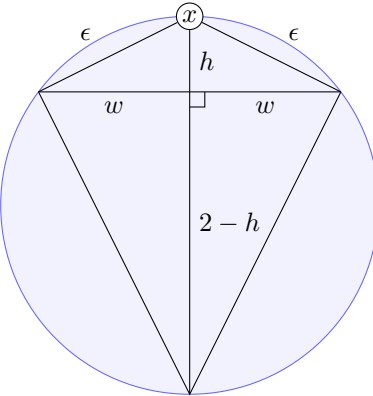

Figure 1: Circular cross-section of the $d$-dimensional unit hypersphere, $\mathcal{S}_d$. The two points $\epsilon$ away from $x$ (in Euclidean distance) and the diametrically opposite point are marked, forming a kite. The diagonals of this kite result in a right triangle of interest to our analysis, with height $h$, width $w$, and hypotenuse $\epsilon$.