# OpenReview forum: "Margin-Independent Online Multiclass Learning via Convex Geometry"
_NeurIPS.cc/2021/Conference — NeurIPS 2021 Poster_

### Official Review · Reviewer_cefD · 2021-07-06

**Rating:** 7
**Confidence:** 4

**Summary:**

The work addresses an online learning problem, where the algorithm receives a sequence of points q_1,q_2,... in the d-dimensional unit ball, and must guess the label of each point among k possible labels {1,...,k}. The correct label corresponds to a nearest-neighbor partition of the ball, R_1,...,R_k determined by k unknown "centers" x_1,...,x_k. If the algorithm predicts that a point q has label i, it incurs a loss equal to the distance between q and the subset R_i of the points labelled as i. The goal is to minimize the total loss. The main result is a set of upper bounds on the loss in the form poly(d), or similar, but in any case independent of the length T of the sequence. The work considers also the special case k=2, the case of general convex classes (for which it gives lower bounds), as well as some other implications of the upper bounds.


**Limitations And Societal Impact:**

The limitations are discussed adequately. There are no foreseeable societal consequences.


**Main Review:**

TLDR: I think this is a good contribution. I believe it could be written in a better way, see below.

Originality: good. The main results are obtained starting from a technique introduced recently, but they are not just a corollary of it. Moreover, different results are obtained through proofs, and some of them require substantial work and a good understanding of the mechanisms behind.

Quality: I did not check the proofs. I would *expect* the quality to be ok based on how the manuscript is written.

Clarity: good. However, I think it that the Introduction can be improved. I would for example pin down the notation more explicitly, since you are studying several variants of the same problem and thus definitions "in words" alone tend to be difficult to follow (the notation is very light and therefore there's room for this). Similarly, I would make the theorem statements more exhaustive, and in particular I would specify the loss under examination (in Theorem 2 the loss is not specified). More in general, I think the results could be structured in a better way; there are many of them, of different kinds, and it's not easy to get the big picture. Otherwise said: what is the main message of this paper?

Significance: fair. The results are significant, as the problem seems basic and the bounds are interesting. I must also say, however, that I find a slight dissonance between the declared goal (margin-independent multiclass learning, see the title) and what the paper actually achieves. First, the "margin independence" can be obtained for any algorithm by simply incorporating the distance into the loss. It is true however that the algorithm presented here subsume the margin-based algorithms in some sense (Thm 5 and Thm 6), so it's not too incorrect to claim "margin independence". Second, the multiclass setting boils down to a very specific scenario: classifiers that are induced by a Voronoi partition. This somewhat clashes with the generality of the first part of the paper.

SPECIFIC COMMENTS:

L48: should you require v to be a unit vector?
1.1: I think that a picture of v,q, and the R_i would help
L68: What do you mean by "we have no control over what feedback we learn about the hidden vector"? In both problems (and in most problems I have seen) the algorithm does not decide what feedback it gets, right?
Theorem 2: you have not specified the loss here. What is it? The same as in 1.1.1 but with the inner product replaces by \delta I guess?


**Time Spent Reviewing:**

6

---

> ### Author Response · Authors · 2021-08-10
> **Response to Reviewer cefD**
>
> Thank you for your thoughtful review of the paper. We agree with your comments regarding the presentation and will take them into account when we update the paper. We respond to some specific comments below:
>
> - L48: Yes, v here should be a unit vector (or at least bounded in norm). We will change this to make this clear.
> - L68: What we mean by “we have no control over what feedback we learn …” is that in many online learning problems (especially those with a “bandits” flavor), the action the learner takes influences what feedback the learner receives (e.g., if the learner pulls a specific arm, the learner only learns the reward for that arm). Contextual search has the property that the feedback the learner receives (whether the learner’s guess was too high or too low) depends on the action of the learner (the learner’s guess), and therefore the learner can tailor their guesses to gain information (and this is somewhat important in the algorithms for contextual search). In the setting we study, the learner does not have this power; regardless of what action the learner takes (the label they assign), they receive the exact same feedback (the true label).
> - Theorem 2: The loss function here (as mentioned in Section 2) should be ell(q, R_i)  = delta(q, x_i) - delta(q, x_i*), where i* is the true label of the query point q. We agree this is confusing and will move this definition up.

---

### Official Review · Reviewer_CoFX · 2021-07-08

**Rating:** 7
**Confidence:** 4

**Summary:**

The paper considers the problem of online prediction of which region of a k-centred nearest-neighbour partition (where “nearest” is defined via a “distance” function) a point is in. The feedback on each trial is which region the point is in - as in standard online learning. However, instead of bounding the number of mistakes, this paper seeks to bound the total loss, where the loss on a trial is the “distance” of the point to the region predicted (see the main review for a confusion I have here). The paper considers two types of “distance” - the (negative of the) inner product and the p-norm. The paper gives loss bounds that are independent of the total number of trials and don’t have a “margin” term.

**Limitations And Societal Impact:**

The algorithm is not robust to noise at all - give the algorithm one incorrect label and it completely breaks.

**Main Review:**

I am surprised that nobody has studied this problem before, especially in the k=2 (separating hyperplane) case - perhaps nobody was able to get anywhere with it until now. If this is truly original then it is an important result and should be accepted. There is one major confusion that I have though - in line 161 what does delta(q,R_i) mean? Surely to be consistent with the abstract it should mean the distance from q to the nearest point in R_i but to be consistent with line 212 it should be the distance from q to the centre of R_i???
Some comments:
Line 77 - you should state here that by “metric” you don’t necessarily mean a metric space (inner product isn’t) and can even be negative.
Line 91 -  “poly(p , d)” should be explicitly written.
Line 211 - around here you drop the subscript t on many qs

**Time Spent Reviewing:**

6

---

> ### Author Response · Authors · 2021-08-10
> **Response to Reviewer CoFX**
>
> Thank you for your thoughtful review of the paper and your detailed comments.
> Re: delta(q, R_i). Thanks for pointing this out; this is a typo, and delta(q, R_i) here should be replaced with delta(q, x_i) (where x_i is the center of R_i). In general, we use delta(x, y) to represent the similarity of a pair of *points* and use ell(x, R) to represent the loss of a point x with respect to a region R, but this is incorrectly used in a few places in section 2 -- we will make sure to fix this.
>
> On a more fundamental level, you are right -- the definition of ell(x, R) we use in the nearest neighbor setting (e.g. difference of Euclidean norms / inner products) is not quite the same thing as “the distance from x to the region R” (which is the definition we use in the convex set setting). We think that for the nearest neighbor formulation of the problem, the former definition is nicer to work with (both from a mathematical point of view and a definitional point of view). Moreover, in many settings, these two definitions are closely related: for example, in the separating hyperplane (k=2) case the two losses differ by at most a multiplicative constant (assuming all vectors are bounded), so any algorithm for one loss also works for the other loss. We will definitely update the paper with a discussion of this point.

---

### Official Review · Reviewer_5eTA · 2021-07-13

**Rating:** 5
**Confidence:** 3

**Summary:**

This paper proposes an online multiclass classifier that minimizes the distance between a point and the correct class partition for that point’s label. They argue that this loss function punishes more harshly points that are classified as extremely wrong versus almost correct. They propose an algorithm where in every round, their classifier predicts a point and suffers the distance based loss between their predicted point and the correct class region.


**Limitations And Societal Impact:**

Limitations: Yes
Negative Societal Impact: N/A, this is a theory paper

**Main Review:**

The paper seems well defined and theoretically sound although I do not think NeurIPS is the correct venue. The paper seems to prove the existence of algorithms with certain properties, but does not elaborate on what the algorithm they are proposing is explicitly. Such a theoretical paper may be better served in a venue such as COLT. It also is clear that there is not nearly enough space in the page limits for this paper.

I do not recommend putting the results section at the beginning of the paper. There is no context for what is discussed there and is extremely confusing until you read the rest of the paper. It also is a waste of space as it is a restatement of theorems that are presented later in the paper. Especially as this paper is already packed with material and space is scarce.

Line 52: what is a time horizon T?

Line 148/149: It is not clear why learning the subsets directly corresponds to being able to label points in an online manner. It is also not clear why the learner is given an adversarially chosen query point in round t; is this point adversarial because it is simply hard to label or the predicted label was wrong in the previous round?

**Time Spent Reviewing:**

2

---

> ### Author Response · Authors · 2021-08-10
> **Response to Reviewer 5eTA**
>
> Thank you for your thoughtful review of the paper. We respond to some of the specific comments below.
>
> - The term “time horizon” is a term from online learning which means the total number of rounds the algorithm runs for (alternatively, the total number of queries the algorithm handles). Regret bounds are often stated as a function of the time horizon (and other relevant parameters).
> - The subset R_i contains the points that should be labelled i, so learning the subsets R_i is in some sense “equivalent” to learning how to label points.
> - The term “adversarial” here (again a term from online learning) simply refers to the fact that the query point in round t can be any possible point (chosen by an “adversary”) -- it has nothing to do specifically with the hardness of labelling or the accuracy of the learner on previous points. This is in contrast to e.g. the points being generated “stochastically” (drawn iid each round from a fixed distribution).

---

### Official Review · Reviewer_vqdM · 2021-07-18

**Rating:** 7
**Confidence:** 3

**Summary:**

The author proposed an online learning setting where the true label is determined by which of k centers is the closest from the query point for a distance function.  The goal of the learner is to minimize the total distance from each query to the region corresponding to the correct label. The authors then propose learning algorithms for these settings and provide a theoretical analysis of the proposed algorithms. The authors also show interesting properties of this learning setting that could not be achieved in the traditional multiclass learning setting.

**Limitations And Societal Impact:**

Yes

**Main Review:**

**Strengths**
1. The new learning setting that proposed by the author is interesting. The framework allows the learner to quantify how wrong the prediction is based on the geometry of the data.
2. The proposed learning algorithms are backed up with strong theoretical analysis.
3. The theoretical results presented in the paper are interesting.
4. The authors provide a thorough theoretical analysis of the learning setting and the proposed algorithms.

**Weaknesses, Concerns, and Questions**
1. The paper contains a lot of new concepts and theoretical analyses. However, the presentation makes it hard to understand.
2. The author presented a new learning setting. Although the learning setting is theoretically interesting, I suggest the author explain more about the motivation of the setting, for example, by providing a concrete example of applications where the learning setting may be useful.
3. Since it contains many new concepts, I suggest the authors explain more the connection of the new concepts with the well knows terminologies in the traditional multiclass setting.
4. Additionally, some illustrations will be useful to help the reader understand the concept and the proposed algorithm, particularly since it is promoted as a geometrically-based method.

In conclusion, I think the proposed learning setting and algorithms are in the interest of the machine learning community. The results and analyses are also interesting. However, I would suggest the author to make the presentation easier to understand by the reader as I mentioned above.



**Time Spent Reviewing:**

4

---

> ### Author Response · Authors · 2021-08-10
> **Response to Reviewer vqdM**
>
> Thank you for your thoughtful review on the paper. We agree with the comments re: presentation and will update the paper to make the presentation easier to understand. Re: motivation, we believe that the algorithms we introduce are relevant in any of the existing settings where online multiclass classification is useful.

---

### Decision · Program_Chairs · 2021-09-27

**Decision:**

Accept (Poster)

**Comment:**

The reviewers are in consensus after discussion that this paper should be accepted even if the score of one reviewer does not reflect this. It is strongly suggested that the presentation around the Introduction be updated (see reviews) and also to make sure that terms with technical meanings such as adversarial are defined for the general reader.